# How are estimated cellular turnover rates influenced by the dynamics of a source population?

**Arpit C. Swain**[1, 2¤]*, **José A. M. Borghans**[2], **Rob J. de Boer**[1]*

**1** Theoretical Biology, Utrecht University, Utrecht, The Netherlands, **2** Center for Translational Immunology, University Medical Center Utrecht, Utrecht, The Netherlands

¤ Current address: Department of Pathology and Cell Biology, Columbia University Irving Medical Center, New York, New York, United States of America
* swainarpit@gmail.com (ACS); r.j.deboer@uu.nl (RJB)

## Abstract

Estimating production and loss rates of cell populations is essential but difficult. The current state-of-the-art method to estimate these rates involves mathematical modelling of deuterium labelling experiments. Current models typically assume that the labelling in the precursors of the population of interest (POI) is proportional to the deuterium enrichment in body water/glucose. This assumption is not always true and it is known that this can have a significant effect on the rates estimated from labelling experiments. Here we quantify the effect that different turnover (replacement) rates of the precursors can have on the estimated proliferation and loss rates of a POI by explicitly modelling the dynamics of the precursors. We first confirm earlier results that the labelling curve of the POI only reflects its own turnover rate if either the turnover rate of the precursors is sufficiently fast, or the contribution from the precursors is sufficiently small. Next, we describe three realistic scenarios with a slowly turning over precursor population, and show how this changes the interpretation of the different parameter estimates. Our analyses underpin that uniquely identifying the turnover rate of a POI requires measurements (or knowledge) on the turnover of its immediate precursors.

## Author summary

The expected lifespan of different types of cells is a fundamental biological quantity that is essential to know in order to advance translational medicine, for example, to improve immune therapies. Deuterium labelling is one of the most reliable techniques used in mice and the only safe technique used in humans to deduce cellular lifespans. Estimating the lifespan of a population of interest (POI) from deuterium labelling data needs mathematical modelling. Most models focus on the POI and do not consider the dynamics of its precursors. Recent reports have

**Data availability statement:** R scripts that were used to make the main and supplementary figures are available at https://github.com/swainarpit/ExplicitSourceChain

**Funding:** ACS was supported by grant number ALWOP.265 of the Dutch Research Council (NWO) to RJdB and by Vici-grant number 09150181910016 of the Dutch Research Council (NWO) to JAMB. The funders had no role in study design, data collection and analysis, decision to publish, or preparation of the manuscript.

**Competing interests:** The authors have declared that no competing interests exist.

shown that this could lead to large errors in lifespan estimates. Our work builds upon this finding and addresses two questions: 1) what quantities can reliably be deduced from deuterium labelling curves?, and 2) which existing estimates need to be re-analysed? We show that the only reliable cases are where the turnover of the POI is much slower than that of its precursors (for example, slowly dividing naive T cells being replaced by rapidly dividing thymocytes). For all other cases, lifespan estimates should be carefully re-evaluated by taking into account information about the precursor populations.

## Introduction

The estimation of the average production and loss rates of a cell population, and therefore of the expected lifespan of its cells, has remained challenging. Most estimation procedures either interfere with the true dynamics of the populations (as in adoptive transfer experiments or in-vitro cultures) or are toxic for the cells (for example during in-vivo labelling with compounds such as BrdU or radioactive diisopropyl-fluorophosphate (DF$^{32}$P)) [1,2]. In recent decades, naturally occurring heavy isotopes (such as $^2$H, $^{13}$C) have been proposed as alternative labelling compounds that are stable and non-toxic, and that should not alter the behaviour of the cells [3,4].

The current state-of-the-art technique to quantify the dynamics of cell populations involves the use of deuterium ($^2$H) as the labelling agent. Deuterated water or glucose is ingested by human volunteers or mice and its incorporation in the DNA of a population of interest (POI) is followed over time. The kinetics of label uptake (upon labelling) and loss (after labelling is stopped) from the DNA of the cells of interest reveals information about the turnover of the POI, which is deciphered using mathematical models [4,5]. These models often simplify all cellular processes into just gain and loss of cells in the POI [5].

It is common practice, in studies addressing maintenance mechanisms of the POI, to distinguish between the gain of cells in the population due to i) cell division and ii) maturation from a precursor population. These are typically considered as two separate processes and are modelled as two independent parameters. Typically, the gain of label from the precursor population has been modelled implicitly, with the assumption that the enrichment of the precursor population mimics that of the body water, i.e., the precursors were assumed to turn over rapidly. In such a case, the rate at which the POI gains label is dictated by its own turnover rate [6]. Here, we investigate different situations in which the labelling dynamics of the source (precursor population) have a significant effect on the labelling curve of the POI. Depending on the turnover rates of the precursors and the POI, the labelling curve of the POI could reflect its own turnover rate, its precursor's turnover rate, or their combination. Thus, to uniquely identify the kinetics of a POI that is (partly) maintained by a source, we emphasize the need to always consider (through prior knowledge or additional measurements) the labelling curve of its immediate precursors.

## Mathematical models

### Implicit source (IS) model

Any population is maintained due to gain of new cells (through a source from a precursor or through proliferation) and loss of existing cells (through differentiation into another population, migration or death). Consider a population, $N$, which is at equilibrium (denoted by $\overline{N}$), has a source of $\sigma$ cells/day, a *per capita* proliferation rate, $p$/day and a *per capita* loss rate, $d$/day (equation 1a). The kinetics of the number of cells, $N$, the number of labelled DNA fragments, $L$, and fraction of labelled DNA, $I$, within this population can be described as:

$$\frac{dN(t)}{dt} = \sigma + (p - d)N(t) \tag{1a}$$

$$\frac{dL(t)}{dt} = \sigma D(t) + pD(t)\overline{N} - dL(t) \tag{1b}$$

$$\frac{dI(t)}{dt} = d(D(t) - I(t)) \tag{1c}$$

where

$$\overline{N} = \frac{\sigma}{(d - p)} \tag{1d}$$

$$D(t) = cU(t) = \begin{cases} 1, & t < 1 \\ 0, & t \geq 1 \end{cases} \tag{1e}$$

This model suggests that the fraction of labelled DNA can be accurately tracked if just the loss rate, $d$, of the cells in the population is known (equation 1c). The turnover rate, $d$, is typically identifiable, as the number of parameters to be estimated reduces from 3 to only 1, when dealing with the labelled fraction of DNA in a cell population at steady state (equation 1a vs equation 1c).

Importantly, in the equations above, the precursor population (source) is assumed to label and de-label instantaneously (equation 1b), i.e., the DNA of the cells arriving from the precursors ($\sigma$) is assumed to follow the fraction of label in the body.

In the model, $D(t)$ is a function denoting the probability that deuterium is incorporated into newly synthesized deoxyribose molecules ($U(t)$ is the level of deuterated water or glucose in plasma (or urine), and $c$ is the intracellular amplification or dilution factor [7]). For simplicity, we treat $D(t)$ as a square pulse where the maximum is scaled to 1 so that the labelled fraction approaches 1 ($I(t) \rightarrow 1$) for very long labelling experiments ($t \rightarrow \infty$). This simplifying assumption does not change the conclusions drawn in this study because the turnover of water and glucose is typically faster than that of the cells in the POI [4,8]. We also scale time such that the labelling period (typically denoted as $\tau$ days) is 1 time unit. Scaling time scales the rates of the population. For example, if the memory T-cell pool in mice (i.e., the POI) is labelled for 28 days and has an estimated turnover rate of 0.02/day, the labelling period becomes 1 (scaled unit of time, reflecting about a month) and the turnover rate of the POI becomes 0.02 x 28 = 0.56 per scaled unit of time (i.e., approximately per month). This scaling simplifies the presentation of the results considerably.

### Explicit source (ES) model

While the previous model assumed that the precursor population labels and de-labels as fast as the body deuterium concentration, in fact, the dynamics of the source population itself can have a major impact. This was illustrated by a study on the dynamics of neutrophils. The lifespan of blood neutrophils was estimated to be ~5 days when the precursors of

neutrophils (in the bone marrow) were assumed to have a fast turnover, and ~0.5 day when the precursors were assumed to be slow, underlining the importance of explicitly considering the dynamics of the source [4,9].

To quantify the effect of the source's dynamics on the rate estimates of the POI, consider an experiment designed to find out the rates of a POI, $N_2$, that has an unobserved precursor population, $N_1$. The cells of the precursors, $N_1$, have a source of $\sigma$ cells/scaled time unit (stu), from their immediate precursors, they divide at a rate of $p_1$/stu and are lost at a rate of $d_1$/stu. A fraction $\alpha$ of the cells leaving the precursor population mature into cells of the POI (i.e., the remaining $(1-\alpha)$ fraction die or move elsewhere). Thus, the precursors obey the standard IS model (equation 1a). If the cells of the POI divide at a rate of $p_2$/stu, and are lost at a rate of $d_2$/stu, the kinetics of the precursors and the POI are written as:

$$\frac{dN_1(t)}{dt} = \sigma + (p_1 - d_1)N_1(t) \tag{2a}$$

$$\frac{dN_2(t)}{dt} = k\alpha d_1 N_1(t) + (p_2 - d_2)N_2(t) \tag{2b}$$

with,

$$\overline{N_1} = \frac{\sigma}{(d_1 - p_1)} \tag{2c}$$

$$\overline{N_2} = \frac{k\alpha d_1 \overline{N_1}}{(d_2 - p_2)} \tag{2d}$$

where $0 < \alpha \leq 1$ and, $\overline{N_1}$ and $\overline{N_2}$ are the steady state values of the precursors and the POI, respectively. If the differentiation of the precursors into the POI is accompanied by one cell division, the parameter $k = 2$, otherwise $k = 1$. See the companion article [10] for a discussion on larger values of $k$. The POI is primarily maintained by the source whenever $k\alpha d_1 \overline{N_1} > p_2 \overline{N_2}$ in equation 2b, which simplifies to the quite natural condition $p_2 < d_2/2$. For simplicity, we no longer mention the units (stu) of the parameters as they are all scaled with respect to the labelling period.

The number of labelled DNA strands in the precursors, $L_1$, and in the POI, $L_2$, in the ES model obey:

$$\frac{dL_1(t)}{dt} = \sigma D(t) + p_1 D(t)\overline{N_1} - d_1 L_1(t) \tag{2e}$$

$$\frac{dL_2(t)}{dt} = \alpha d_1 L_1(t) + (k-1)\alpha d_1 \overline{N_1} D(t) + p_2 D(t)\overline{N_2} - d_2 L_2(t) \tag{2f}$$

Thus, if differentiation is accompanied by cell division, the POI gains label due to the inflow of labelled DNA strands from the precursors as well as due to the generation of de-novo DNA strands on replication of the labelled and unlabelled DNA strands. The kinetics of the fractions of labelled DNA strands in the precursors, $l_1$, and the POI, $l_2$, then follow:

$$\frac{dl_1(t)}{dt} = d_1 (D(t) - l_1(t)) \tag{2g}$$

$$\frac{dl_2(t)}{dt} = \frac{(d_2 - p_2)}{k}(l_1(t) + (k-1)D(t)) + p_2 D(t) - d_2 l_2(t) \tag{2h}$$

$$= \frac{d_2}{k}(l_1(t) + (k-1)D(t) - kl_2(t)) + \frac{p_2}{k}(D(t) - l_1(t)) \tag{2i}$$

Transforming the system from cell numbers into fractions reduces the system to a 3-parameter model: $d_1$, $p_2$ and $d_2$ (equation 2i). Note that after a long labelling period (or in rapidly turning over populations), $l_1(t) \to D(t) = 1$, and that regardless of the value of $k$, $l_2(t) \to 1$.

In practice, the POI is often measured in blood and tends to be at the end of a series of cell populations. For example, neutrophils in the blood are preceded by several cell populations in the bone marrow, e.g., mature neutrophils, banded neutrophils, metamyelocytes, myelocytes, promyelocytes, myeloblasts, and all the way up until haematopoietic stem cells. Fortunately, the ES model shows that simply measuring the fraction of labelled precursors (and not even knowing their true rates) is enough to estimate the true rates of the POI (equation 2h). Thus, one does not need to know the labelling dynamics of all the sub-populations in the entire chain from the precursor populations to the POI to find an accurate estimate of the POI's *per capita* rates.

## Results

### The ES model can perfectly describe the dynamics of a two sub-population kinetically heterogeneous population

We generalized the IS model above by simply introducing a source population in the ES model which allows us to model either a slow or a fast source population. The POI in the ES model, however, is still modelled as a homogeneous population. Yet, the ES model can describe a two sub-population kinetically heterogeneous population perfectly because the two sub-population kinetic heterogeneity model [6,11] turns out to be a special case of the ES model. To show this, we compare the solutions of the two sub-population kinetically heterogeneous model (Equation 3) and the ES model (Equation 4).

The solution of the two sub-population kinetically heterogeneous model [11] is:

$$I(t) = \beta \left(1 - e^{-\delta_1 t}\right) + (1 - \beta)\left(1 - e^{-\delta_2 t}\right)$$

(3)

where $\delta_1$ and $\delta_2$ are the turnover rates of the first (slow) and the second (fast) sub-populations, respectively, because we define $\beta \in [0, 1]$ for the fraction of the population that is made up of the slow sub-population (without loss of generality).

During the labelling phase, the solution of the ES model is:

$$l_1(t) = 1 - e^{-d_1 t}$$

(4a)

$$l_2(t) = 1 - \frac{(d_2 - p_2)}{(d_2 - d_1)} e^{-d_1 t} - \frac{(p_2 - d_1)}{(d_2 - d_1)} e^{-d_2 t}$$

(4b)

$$= \alpha \left(1 - e^{-d_1 t}\right) + (1 - \alpha)\left(1 - e^{-d_2 t}\right)$$

(4c)

where $\alpha = \frac{(d_2 - p_2)}{(d_2 - d_1)} \in (-\infty, +\infty)$. We, here, consider the $k = 1$ case of the ES model because the $k = 1$ case is the more general model and includes the $k = 2$ case (see the sub-section ***division-linked differentiation*** below). The $k > 2$ case is studied in the companion paper [10].

The solutions of the POI in the ES model (equation 4c) and the two sub-population kinetically heterogeneous model (Equation 3) are identical. As $\alpha \in (-\infty, +\infty)$ has a wider range than $\beta \in [0, 1]$, the ES model can perfectly describe any two sub-population kinetically heterogeneous population but the converse is not always true. Further, if $k = 2$ and $p_1 > 0$, the ES model can also represent a population of stem cells, $N_1$, dividing asymmetrically into themselves ($N_1$) and into the following population ($N_2$) (see equation A in S1 Text). Thus, the ES model is the more general model and can provide alternate explanations of many other mechanistic models.

Although this equality makes it difficult to recognize the true mechanism underlying a labelling curve, the ranges of $\alpha$ and $\beta$ help us identify scenarios where the mechanism is unambiguous. To find scenarios where the dynamics of the ES model is identical to that of the two sub-population kinetic heterogeneity model, we compare the relationship between the

parameters of the ES and the two sub-population kinetic heterogeneity models. As $\beta > 0$, we only consider $\alpha > 0$, meaning that $d_2 > d_1$, i.e., that the POI is faster than its source population. Comparing Equations 3 and 4c, we find that $\alpha = \beta$, and

$$\alpha = \frac{(d_2 - p_2)}{(d_2 - d_1)} \tag{5}$$

$$\Rightarrow p_2 = \alpha d_1 + (1 - \alpha)d_2 = \beta \delta_1 + (1 - \beta)\delta_2 = \overline{\delta}$$

when $d_1 = \delta_1$ and $d_2 = \delta_2$. That is, $\alpha$ in the ES model gives the fraction of the two sub-population kinetically heterogeneous population that is made up of the slow sub-population, and the proliferation rate of the POI, $p_2$, in the ES model is the same as the average turnover rate, $\overline{\delta}$, estimated for the two sub-population kinetically heterogeneous population. For example, deuterium labelling of CD8$^+$ memory T cells [12] can be described equally well with either the two sub-population kinetic heterogeneity model (as was shown by Westera et al. [12]) or the ES model. Additional information can sometimes help to determine the choice of the model. For example, as memory T cells specific to LCMV-Armstrong are maintained by homoeostatic divisions [13], one can confidently choose a two sub-population kinetic heterogeneity model without a source over the ES model when evaluating their labelling data.

As the labelling curve of POI in both models is defined as the sum of two exponentials, the turnover rates of the precursor and the POI in the ES model have to be the same as those of the slow and fast sub-populations in the two sub-population kinetic heterogeneity model, respectively. Therefore, based on its deuterium labelling dynamics, a kinetically homogeneous population with a source cannot be distinguished from a two sub-population kinetically heterogeneous population whenever the POI is faster than its source population (i.e., when $d_2 > d_1$, meaning $\alpha > 0$). In cases where $d_2 < d_1$, one can, therefore, be certain that the population is not kinetically heterogeneous.

**The labelling curve of the POI reflects its own or its precursor's turnover rates in special cases**

In the ES model, the POI can have a source (of potentially recently divided cells), can divide, and has a loss rate. Trivially, the labelling curve of the ES model is defined by the turnover rate of the POI, $d_2$, if the source into the POI is negligible, i.e., if $p_2 \to d_2$ (in equation 2i). Therefore, in this section, we discuss the label gain and loss rates in two special cases of the ES model where the POI does have a significant source from the precursors: 1) a rapidly turning over precursor, and 2) a rapidly turning over, non-dividing POI. Throughout this article, we use label gain rate (and label loss rate) to refer to the rate at which a population gains (and loses) labelled DNA.

If the precursors turn over rapidly, e.g., $d_1 \gg 1$, the unlabelled cells in the precursor population would be rapidly replaced by labelled cells. The precursors can, in such a case, be approximated by the deuterium availability in plasma, i.e., $I_1(t) \approx D(t)$ (equation 2g). The labelling in the POI (equation 2i) will then boil down to:

$$\frac{dI_2(t)}{dt} \approx d_2 \left( D(t) - I_2(t) \right) \tag{6}$$

which is the familiar IS model, showing that the enrichment of the POI is determined by just the turnover rate of the POI, $d_2$. Thus, the gain and loss rates estimated from labelling experiments where the precursors turn over rapidly (relative to the labelling period) indeed reflect the turnover rate of the POI [6]. For example, as rapidly dividing thymocytes are the precursors of slowly dividing naive T cells, the estimated lifespan should correctly reflect the true lifespan of naive T cells [14]. See Box 1 for scenarios when the estimated rates are accurate.

If the only source of label into the POI is due to the flow of labelled DNA from the precursors (i.e., $k = 1$ and $p_2 = 0$), the labelling dynamics in the POI (equation 2i) obeys:

$$\frac{dl_2(t)}{dt} = d_2 \left( l_1(t) - l_2(t) \right) \tag{7}$$

If the cells of the POI turn over rapidly (e.g., $d_2 \gg 1$), the enrichment of the POI mimics that of the precursors (Equation 7). Therefore, measuring only the POI will, in fact, reveal the turnover rate of the precursors. The neutrophils are a prime example of this case, as it is unlikely that the slow labelling rate of mature neutrophils in blood reflects their own rate. Their rate of label accrual probably reflects the slow turnover of their precursors in the bone marrow [4,9]. Note that when $k > 2$, most precursors become labelled upon arriving in the POI (see the companion paper by *Yan et al.* [10]).

The special cases can also be derived from the function describing the fraction of labelled DNA strands in the POI (equation 4b). When either the precursors or the POI are extremely fast, the fraction of labelled DNA strands in the POI boils down to $\lim_{d_1 \to \infty} l_2(t) = 1 - e^{-d_2 t}$ or $\lim_{d_2 \to \infty} l_2(t) = 1 - (1 - \alpha_2)e^{-d_1 t}$, which is only dependent on either the turnover of the POI or that of the precursors, respectively. Here, $\alpha_2 = \frac{p_2}{d_2}$ is the fraction of the POI that is replaced by cell division. These special cases showcase that the labelling curve of the POI sometimes reflects the turnover rate of the POI and sometimes that of its precursors.

**The label gain and loss rates of the POI do not equal its turnover rate if the POI and the precursors have comparable turnover rates**

Above we estimated the label gain and loss rates in a few extreme cases where either the source into the POI is negligible, or the precursors or the POI are very short-lived (i.e., the precursors or the POI become labelled very rapidly). In many labelling experiments, the duration of the labelling period is comparable to the POI's lifespan, to avoid scenarios where the POI either hardly labels or becomes labelled very rapidly during the labelling period. This is important as an accurate estimation of the POI's rates requires the collection of enough informative data points during the labelling period. As estimating the label gain and loss rates in these cases is not straight-forward, we derive first-order approximations for the POI's label gain and loss rates.

Denoting the label gain and loss rates as $p^*(t)$ and $d^*(t)$, respectively, the labelling curve of a POI can be written as (i.e., generalizing equation 1c using time-varying parameters):

$$\frac{dl(t)}{dt} = p^*(t)D(t) - d^*(t)l(t) \tag{8}$$

The label gain rate reflects the dynamics of the entire population [5], as de-novo labelled DNA molecules are made if a cell (with either labelled or unlabelled DNA molecules) divides during the labelling phase. The label loss rate, however, reflects the loss rate of the labelled sub-population only [5], as the POI loses label only when a cell with labelled DNA is replaced by a cell with unlabelled DNA.

In the ES model introduced above, the fraction labelled DNA in the precursors is determined only by its loss rate, $d_1$, whereas the fraction labelled DNA in the POI is dependent not just on its own rates but also on the rates of its precursor population (equation 2h). The rate of label gain in the POI, $p^*(t)$, during the labelling phase ($t \leq 1$, which means $D(t) = 1$) is (from equations 2h and 4b):

$$p^*(t) = \frac{dl_2(t)}{dt} = \frac{(d_2 - p_2)}{k}(l_1(t) + k - 1) + p_2 - d_2 l_2(t) \tag{9a}$$

$$= \begin{cases} d_2 e^{-d_2 t} - \dfrac{(d_2 - p_2)}{k(d_2 - d_1)}(d_2 e^{-d_2 t} - d_1 e^{-d_1 t}), & | \ d_1 \neq d_2 \\[2ex] d_2 e^{-d_2 t} - \dfrac{(d_2 - p_2)(1 - d_2 t)}{k} e^{-d_2 t}, & | \ d_1 = d_2 \end{cases} \tag{9b}$$

The rate of label gain depends on both the precursor and the POI, as $p^*(t)$ also depends on $I_1(t)$. The initial label gain rate of the POI (i.e., when $t = 0$) is:

$$p^*(0) = \frac{d_2(k-1) + p_2}{k} \tag{9c}$$

The initial label gain rate, $p^*(t)$, can be much slower than the true turnover rate of the POI, $d_2$, if the precursors are slow (see the previous section on special cases). Therefore, it might be beneficial for our analysis to express the labelling in the POI as a function of the label gain rate, $p^*(t)$, instead of the true turnover rate of the POI. To calculate the rate of label gain later during the labelling phase, the expression of label gain rate (equation 9a) needs to be further simplified. If the label gain can be approximated well by a linear increase (i.e., if the turnover rates of the precursors and the POI are small relative to the labelling period, which is the case for well-designed experiments), the expression for the gain rate (equation 9a) can be simplified by noticing that $I_1(t) = 1 - e^{-d_1 t} \approx d_1 t$, and $I_2(t) \approx p^*(t)t$. During the labelling phase ($t < 1$), the rate of label gain in the POI can then be approximated by:

$$p^*(t) \approx \frac{d_2\,(d_1 t + k - 1) + p_2(1 - d_1 t)}{k(1 + d_2 t)} \tag{9d}$$

The approximate rate of label gain at the end of the labelling phase (when $t \to 1$) of the POI, then, becomes

$$p^*(1) \approx \frac{d_2\,(d_1 + k - 1) + p_2(1 - d_1)}{k(1 + d_2)} \tag{9e}$$

If we are in a regime where the gain of label in the POI can be approximated well by a straight line, these expressions become similar, i.e., $p^*(t) \approx p^*(0) \approx p^*(1)$ (see Figs 1 and 2, and Table 1 below).

Additionally, the above expressions can be used to determine the behaviour of the precursors relative to the POI. If the labelling curves of both the precursors and the POI can be approximated reasonably by a straight line, the population with the faster labelling dynamics can be identified just by comparing the initial gain rate of these populations. Hence, the gain in label in the POI (during the labelling phase) is faster than that in its precursor if

$$I_2(t) > I_1(t) \Leftrightarrow p^*(0)t > d_1 t \Leftrightarrow p^*(0) > d_1 \tag{10}$$

As opposed to the rate of label gain, $p^*(t) = \frac{dI_2(t)}{dt}$, which is defined on the total population, the label loss rate, $d^*(t) = \frac{1}{I_2(t)}\frac{dI_2(t)}{dt}$, is defined on the labelled fraction (see Equation 8). So, the rate of label loss in the POI, $d^*(t)$, in the de-labelling phase (i.e., when $D(t) = 0$ in equation 2h) is:

$$d^*(t) = -\frac{\frac{(d_2 - p_2)}{k}I_1(t) - d_2 I_2(t)}{I_2(t)} \tag{11a}$$

$$= d_2 - \frac{(d_2 - p_2)}{k}\frac{I_1(t)}{I_2(t)} \tag{11b}$$

Again, as $I_1(t) \approx d_1 t$ and $I_2(t) \approx p^*(t)t$, the above expression simplifies to

$$d^*(t) \approx d_2 - \frac{(d_2 - p_2)}{k}\frac{d_1}{p^*(t)} \tag{11c}$$

It is important to keep in mind that the approximated downslope is accurate for only a short period of time in the de-labelling phase as the approximations of $I_1(t)$ and $I_2(t)$ are not accurate for $t > 1$.

PLOS Computational Biology

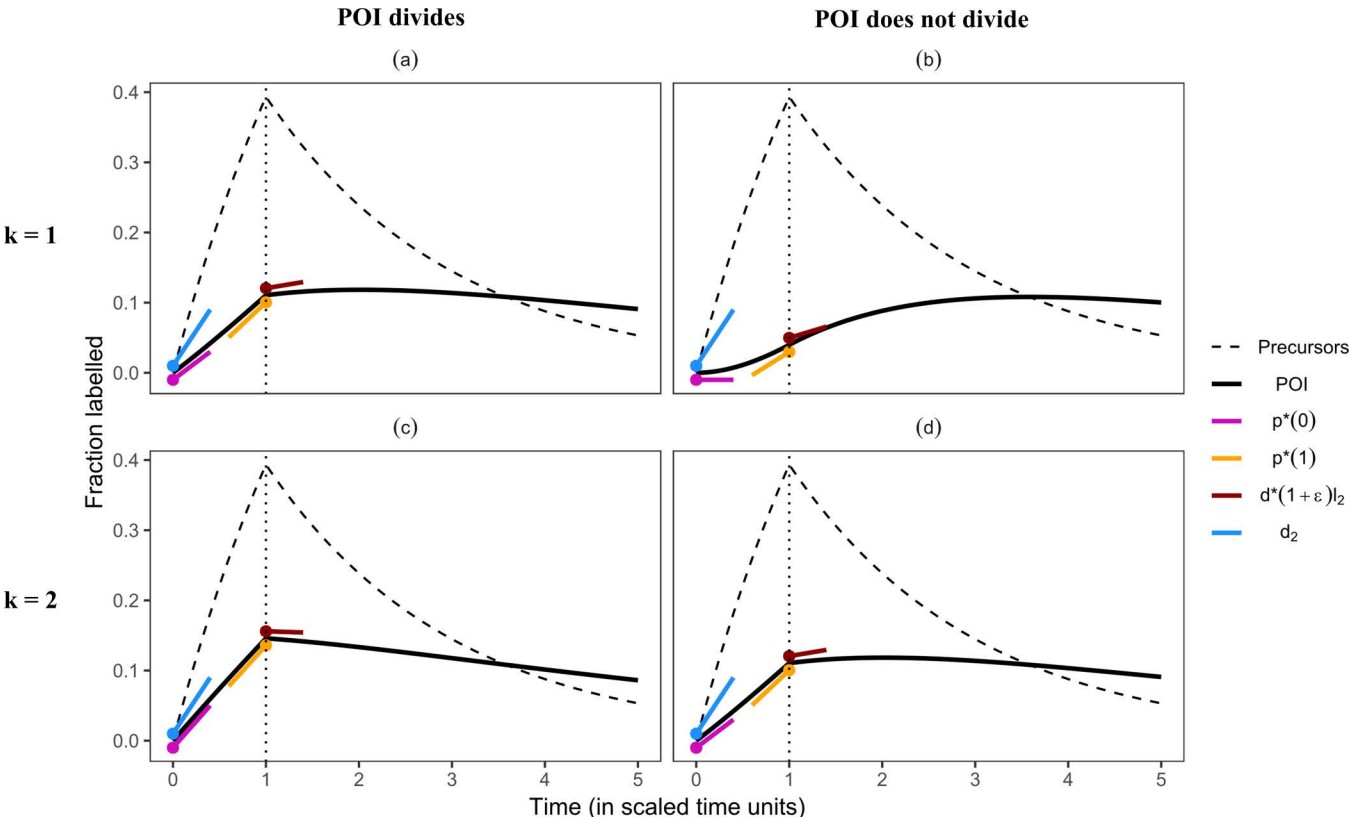

**Fig 1. The rate of label gain in the POI is lower than the POI's turnover rate, $d_2$, if the precursors turn over faster than the POI.** The filled circles mark the timepoints at which the slopes were calculated (note that unlike $p^*(t)$ and $d^*(t)$, $d_2$ is constant over time). A small offset was added to the slopes for better visualization. The slopes denoting initial, $p^*(0)$ (equation 9c), and final, $p^*(1)$ (equation 9e), label gain rates of the POI were given a small negative offset, while those denoting the true turnover rate, $d_2$, and the initial label loss rate, $d^*(1+\in)l_2$ (equation 11d), of the POI were given a small positive offset. The slopes were calculated as the label gain or loss rate multiplied by the total, i.e., one, or the fraction of labelled cells (i.e., $l_2$), respectively, for example, $d^*(1+\in)l_2$. The loss rates of the precursors, $d_1$, and of the POI, $d_2$, were set to 0.5 and 0.2, respectively. The division rate of the POI, $p_2$, was either set to 0.1 (in (a) and (c)) or to 0 (in (b) and (d)). Note that the dynamics in (a) and (d) are identical (see the sub-section ***Division-linked differentiation*** below for the explanation). In these simulations, the body deuterium concentration was described as a step function (equation 1e).

As $p^*(0) = 0$ in some cases, for example when $p_2 = 0$ and $k = 1$, we use the approximation $p^*(t) \approx p^*(1)$ for defining $d^*(t)$. Thus, the initial rate at which label is lost in the beginning of the de-labelling phase is

$$d^*(1+\varepsilon) \approx d_2 - \frac{(d_2 - p_2)}{k} \frac{d_1}{p^*(1)}$$

(11d)

where $\varepsilon > 0$ is small.

The approximations of the gain and loss rates of label in the POI show that these rates are not dictated only by the turnover rate of the POI but also by the division rate of the POI and the turnover rate of the precursors. Next, we use the above simplified expressions for the gain and loss rate of label in the POI to find qualitative relationships between the estimated and true label gain or loss rates.

## The rate of label uptake in the POI is generally lower than its true turnover rate

Models that consider an implicit source (like the IS model above in Equation 1 or the implicit kinetic heterogeneity model [5]) predict that, in a population that is at steady state, the rate of label gain in the POI, $p^*(t)$, represents the average

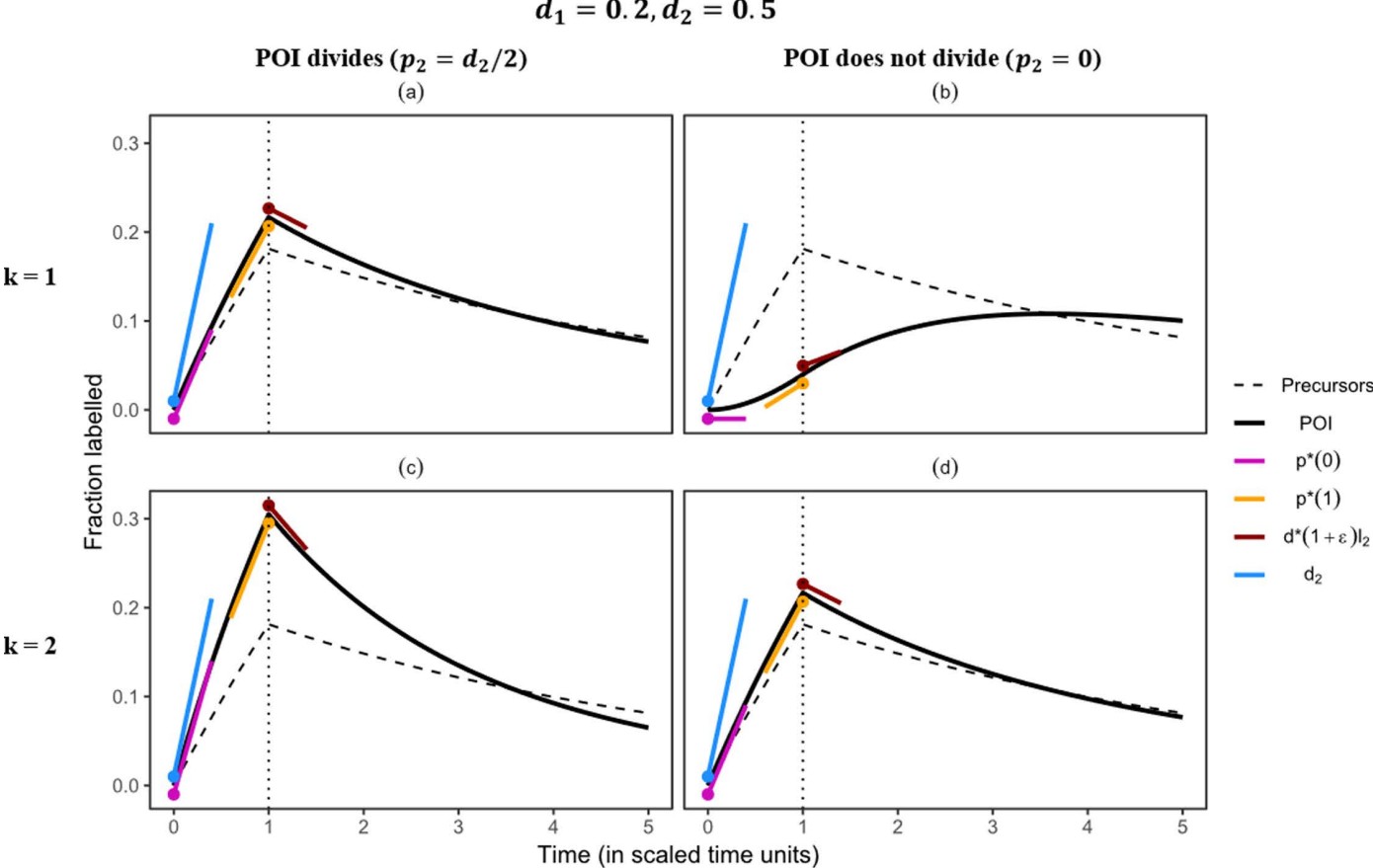

**Fig 2. The rate of label gain in the POI is also lower than the POI's true turnover rate, $d_2$, if the precursors turn over more slowly than the POI.** The filled circles mark the timepoints at which the slopes were calculated (note that unlike $p^*(t)$ and $d^*(t)$, $d_2$ is constant over time). A small offset was added to the slopes for better visualization. The slopes denoting the initial label gain rate, $p^*(0)$ (equation 9c), and the final label gain rate, $p^*(1)$ (equation 9e), of the POI were given a small negative offset, while those denoting the true turnover rate, $d_2$, and the initial label loss rate, $d^*(1+\epsilon)l_2$ (equation 11d), of the POI were given a small positive offset. The slopes were calculated as the label gain or loss rate multiplied by the total, i.e., one, or the fraction of labelled cells (i.e., $l_2$), respectively, for example, $d^*(1+\epsilon)l_2$. The loss rate of the precursors, $d_1$, and the POI, $d_2$, were set to 0.2 and 0.5, respectively. The division rate of the POI, $p_2$, was either set to 0.25 (in (a) and (c)) or to 0 (in (b) and (d)). Note that the dynamics in (a) and (d) are identical, which can also be seen in Table 1 (see the sub-section **Division-linked differentiation** below for details). In these simulations, the body deuterium concentration was described as a step function (equation 1e). Note the large difference in the labelling curves of a non-dividing POI depending on whether differentiation is accompanied by cell division (panels (b) and (d)).

turnover rate of the POI ($d$ in the IS model and $p^*$ in the implicit kinetic heterogeneity model [5]). To test the validity of this prediction, we graphically compare the label gain and loss rates of the POI found from the analytical predictions (equations 9c–e and 11c) and from numerical estimations, using the implicit kinetic heterogeneity model (Equation 8 with $p^*$ and $d^*$ instead of $p^*(t)$ and $d^*(t)$, respectively) with actual labelling curves of the ES model (Figs 1, 2 and Fig A in S1 Text and Table 1) in different scenarios. Similar conclusions can be drawn when $D(t)$ is not modelled as a square pulse, but instead reflects the slower gain and loss of deuterium enrichment in body water that is typically observed during deuterated water labelling experiments (Figs B-D in S1 Text).

**Case I: When the turnover rates of the precursors and the POI are comparable.** We first consider scenarios where the lifespans of the precursors and the POI are comparable to the duration of the labelling period (e.g., $0.1 \leq d_1 < 1$, $0.1 \leq d_2 < 1$), for example, labelling studies that track memory T cells, which might have a source from less-differentiated

**Table 1. The true proliferation and turnover rate ($d_1$, $d_2$, and $p_2$), and the predicted label gain and loss rates (i.e., $p^*(0)$, $p^*(1)$, and $d^*(1+\in)$) using Equations 9c, 9e, sand 11d) corresponding to the simulations shown in**Figs 1 and 2**. The parameters ($p^*$ and $d^*$) of the best fits (shown in Figs E and F in S1 Text) of the phenomenological model (Equation 8) to the labelling data of the POI. The estimated label gain rate initially (see $p^*(0)$) reflects the proliferation rate of the POI (when $k=1$) and then changes to reflect the influence of the precursor's rate (captured by $p^*(1)$). The estimates of $p^*$ and $d^*$ are not reported for the cases in which the phenomenological model fails to describe the data (see Figs E and F in S1 Text). Note that the estimated loss rate, $d^*(1+\in)$, gives the rate at which the labelled population loses labelled DNA soon after the end of the labelling period ($\in$ is a very small positive number). The negative sign indicates that the population, in fact, gains labelled DNA for a short period after the end of labelling (see Equation 8). The rates (expressed as stu) are scaled with respect to the labelling period.**

| Condition | $d_1$ | $d_2$ | $p_2$ | $k$ | $p^*(0)$ | $p^*(1)$ | $d^*(1+\in)$ | $p^*$ | $d^*$ |
|---|---|---|---|---|---|---|---|---|---|
| Faster precursors | 0.5 | 0.2 | 0.1 | 1 | 0.10 | 0.13 | -0.20 | 0.12 | 0.04 |
| | | | | 2 | 0.15 | 0.15 | 0.03 | 0.16 | 0.13 |
| | | | 0 | 1 | 0 | 0.08 | -1 | – | – |
| | | | | 2 | 0.10 | 0.13 | -0.20 | 0.12 | 0.04 |
| Slower precursors | 0.2 | 0.5 | 0.25 | 1 | 0.25 | 0.20 | 0.25 | 0.24 | 0.27 |
| | | | | 2 | 0.38 | 0.27 | 0.41 | 0.37 | 0.40 |
| | | | 0 | 1 | 0 | 0.07 | -1 | – | – |
| | | | | 2 | 0.25 | 0.20 | 0.25 | 0.24 | 0.27 |

memory T-cell phenotypes, in blood [15]. The numerical simulations explore examples where the precursors and the POI have comparable lifespans, e.g., the considered lifespans are either 2-fold or 5-fold longer than the duration of the labelling period (Figs 1 and 2). Further, we show cases where the POI may (left column in figures, $p_2 > 0$) or may not (right column in figures, $p_2 = 0$) divide.

***Precursors are faster than the POI.*** If the precursors turn over faster than the POI, i.e., $d_1 > d_2 > p_2$, the labelling in the POI, during the labelling phase, cannot be higher than that in the precursors ($p^*(0) < d_1$ in Equation 10, Fig 1).

If the differentiation of the precursors into the POI is not accompanied by division (i.e., if $k = 1$), the initial rate of label gain in the POI, $p^*(0)$, is given by its division rate, $p_2$ (Equation 9c, Fig 1a–b, and Table 1). As the predicted initial label gain rate, $p^*(0) = p_2$, can have any value between $0$ and $d_2$ (Fig 1a–b), the initial slope is not the average turnover rate of the POI, as was previously assumed. If the cells in the POI do not divide (i.e., if the gain of label in the POI is only due to the influx of labelled precursors), the labelling curve of the POI has an initial delay during which the gain of label in the POI is zero (Fig 1b). This initial delay in the gain of label is given by $p^*(0) = p_2 = 0$, which is infinitely smaller than the expected $d_2$.

After the initial phase, the labelling curve is best described by $p^*(1)$, which depends on all three parameters of the model (equation 9e). The slope, $p^*(t)$, increases over time, starting at the POI's division rate, $p_2$, up to $p^*(1)$, where the gain rate of label in the POI is higher than its own division rate, and is approaching its turnover rate, $d_2$ (Table 1). The slope of the labelling curve approaches the turnover rate of the POI as cells of the POI that are lost are replaced by labelled cells from the higher enriched precursors. Numerical simulations confirm that the predicted slopes describe the true labelling curve faithfully (Fig 1a–b) and are similar to each other if the POI is a dividing population (Fig 1a).

If the differentiation of precursors into the POI is accompanied by cell division, the initial rate of label gain in the POI cannot be zero (i.e., if $k = 2$ in Equation 9c, Fig 1c–d). Like before (the case when $k = 1$), the approximated gain rate increases from the initial $p^*(0)$ to $p^*(1)$ due to the higher enrichment of the incoming precursors. The label gain rate of the POI is closer (relative to the case when $k = 1$) to, but still smaller than, its true turnover rate (Fig 1c–d). Thus, if the precursors are faster than the POI, the label gain rate in the POI increases over the labelling period due to an influx of precursors into the POI, provided $d_1, d_2 < 1$ and regardless of whether differentiation is accompanied by division (Table 1).

***Precursors are slower than the POI.*** If the POI is faster than the precursors, i.e., $d_2 > d_1$, the labelling in the POI can be either faster or slower than the labelling in the precursors (Equation 10, Fig 2a–b, and Table 1). The labelling of the POI is faster than that of the precursors if the division rate of the POI is faster than the turnover rate of the precursors,

i.e., $p_2 > d_1$. If the cells in the POI do not divide, the labelling in the POI can be faster than that in the precursors only if the precursors go through division-linked differentiation and the POI is at least 2-fold faster than the precursors, $d_2 > 2d_1$ ([Equation 10]; also see Equations 14 and [15]). Note that the labelling of the POI lags behind the labelling of its precursors if new cells in the POI are generated without division, either by differentiation or by self-renewal (compare [Fig 2a]–[2d]).

Some properties stay unchanged compared to the case where precursors are faster than the POI. For example, if the precursors do not divide while differentiating into the POI ($k = 1$), the initial rate of label uptake reflects the division rate of the POI, $p^*(0) = p_2$ (equation [9c]). This implies that a non-dividing POI would initially have a zero labelling rate ([Fig 2b]), which should increase over time to reflect the turnover rate of the precursor population. If the precursors go through division-linked differentiation ($k = 2$), the initial rate of label gain in the POI reflects both its division and death rates (equation [9c]). The approximated slopes ($p^*(0)$ and $p^*(1)$) are, in all cases, markedly lower than the true turnover rate of the POI, $d_2$. The slope declines over time during the labelling period, starting at a value defined by its own (division and loss) rates and moving towards the turnover rate of the precursors. The low enrichment of the precursors dilutes the enrichment in the POI, causing a decline in the slope of the labelling curve. Therefore, the POI's label gain rate initially reflects its own division and death rates, and is only influenced by the rates of its precursors later in the labelling period.

Finally, to confirm these analytical results, we defined two constant parameters $p^*$ and $d^*$ instead of $p^*(t)$ and $d^*(t)$, respectively, in the phenomenological implicit kinetic heterogeneity model ([Equation 8]) to numerically estimate the label gain and loss rates of the POI in all cases (i.e., whether the precursors are faster or slower than the POI) by non-linear parameter fitting. In all cases where the model was able to describe the artificial data, the estimated label gain rate, $p^*$, lay between the analytically predicted initial and final label gain rates ($p^*(0)$ and $p^*(1)$, respectively) ([Table 1]). The estimated label loss rate, $d^*$, and the predicted initial label loss rate, $d^*(1+\in)$, however, differed considerably. As the estimated label loss rate, $d^*$, is based upon the labelling dynamics of the POI during the entire de-labelling phase, it need not be similar to the initial gain or loss of label after the end of the labelling period (i.e., $d^*(1+\in)$, [Table 1]). Note that the phenomenological model ([Equation 8]) is unable to describe the labelling curve if the precursor population differentiates without division into a non-dividing POI (see [Table 1] and Figs E and F in S1 Text).

**Case II: The turnover rates of the precursors and the POI are not comparable.** If the POI is much faster than the precursors, the label gain rate approaches the turnover rate of the precursors, which was highlighted above as a well-known special case [4,9]. Note that the approximated slopes describe the labelling curve for a short period of time, as linear approximations do not give good descriptions of $I_1(t)$ and $I_2(t)$ when $d_1, d_2 > 1$ (see Fig A and Table A in S1 Text). Similarly, if the precursors are faster than the POI, the label gain rate in the POI approaches its own turnover rate as the label gain rate in the POI cannot be higher than its turnover rate, which was also reported above as a special case. Therefore, the approximated rates give a good description of the gain and loss of label in a population.

***The label gain rate can maximally be the turnover rate of the POI.*** Intuitively, the label gain rate has to be lower than the true turnover rate of the POI as unlabelled DNA strands cannot be replaced faster than the rate at which the cells of the POI are replaced. The examples above confirmed this intuition. Here we prove that the rate at which the POI gains label is invariably lower than the POI's turnover rate..

The label gain rate, $p^*(t)$, (equation [9b]) can be re-written as:

$$p^*(t) = d_2 \left( \frac{ab}{(1-b)} e^{-d_1 t} + \left( 1 - \frac{a}{(1-b)} \right) e^{-d_2 t} \right) = d_2 q(t)$$

(12a)

where, $a = \frac{(1-\alpha_2)}{k}$, $b = \frac{d_1}{d_2}$ and $\alpha_2 = \frac{p_2}{d_2}$.

If $t_c$ denotes the timepoint where $q(t)$ is at its maximum (see equation B in S1 Text for details), then the upper bound of the label gain rate, $p^*(t)$, is:

$$p^*(t_c) = d_2 q(t_c) = d_2 e^{-d_2 t_c} \left( 1 - \frac{(1-a)}{b} \right) < d_2$$

(12b)

$$\Rightarrow p^*(t) < d_2 \tag{12c}$$

as $\left(1 - \frac{(1-a)}{b}\right) < 1$ when $a < 1$, which is always true. Thus, the rate at which the POI gets labelled is always lower than its turnover rate, $d_2$. Interpreting the label gain rate as the true turnover rate would, therefore, overestimate the POI's lifespan whenever the precursors are not very fast and/or play a significant role in the maintenance of the POI.

### The initial label loss in the POI is accurately predicted

We have shown that the calculated label gain rates predict the enrichment in the precursors and the POI accurately in most cases. As the relative enrichment of populations at the end of the labelling period influences the rate at which label is lost (Equation 11b), the initial rate at which label is lost, $d^*(1)$, can be approximated well by using the estimated gain rates (equation 11c).

If the lifespans of the POI and the precursors are comparable to the labelling period, the estimated gain and loss rates give a good description of the labelling curves (see the red and green arrows in Figs 1 and 2). The estimated loss rates predict whether the POI start to lose or continue to accrue label at the end of the labelling phase (Table 1). If the precursors are more enriched than the POI, the labelling of the POI continues after the labelling period in most cases (4 out of 5 cases, see Figs 1a, 1b, 1d and 2b). In these cases, the gain of label in the POI is primarily due to the inflow of highly labelled precursor cells (also see the subsection *Timing of the peak*). In the other four cases (see Figs 1c, 2a, 2c and 2d), the label gain in the POI was driven by proliferation.

Note that even if the cells in the POI are very short-lived, the estimates of the initial loss rate accurately informed on whether the POI gained or lost label after the labelling period (Table A in S1 Text). So, the estimated rates do provide a faithful description of the de-labelling curve, but only for a very short period of time (Fig A in S1 Text). Thus, the predicted label gain rate, $p^*(1)$, can be used to find a good approximation of the initial loss rate (equation 11c).

### The POI can have four different labelling behaviours

Unlike the IS model where the labelling of the POI is determined only by the death rate of the POI (Equation 1c), in the ES model the POI can have four qualitatively different slopes at which the POI gains and loses label (summarized in Table 2). The labelling of the POI is determined by:

1) its own turnover rate, $d_2$, if the precursors turn over rapidly. In this case, the labelling in the precursors approximates the label availability in the plasma. The ES model approximates the IS model, and the rates of labelling and de-labelling are primarily driven by the POI's loss rate. An example of this case are naive T cells, that have the rapidly dividing thymocytes as their precursors [14]. Note that if $k \gg 2$, the ES model approximates the IS model (see the companion paper by *Yan et al.* [10]).

2) the turnover rate of the precursors, $d_1$, if the POI is fast. An example would be mature neutrophils in blood, which are thought to mimic the labelling curve of their precursors in the bone marrow [4,9]. The POI, here mature neutrophils, are probably much faster than their precursors in the bone marrow. They attain the enrichment of the precursors as they are replaced (by the cells flowing in from the precursors, post-mitotic pool) much faster than the change in the enrichment of the precursors.

3) its division rate, $p_2$, if differentiation is not accompanied by division ($k = 1$). This is because the increase in enrichment due to the division in the POI would initially be the major contributor compared to the label flowing in from the precursors. Note that the precursors and the POI should have lifespans that are comparable to the length of the labelling period. A likely example for such a case would be central memory T cells undergoing homeostatic differentiation into effector memory T cells [16,17].

**Table 2. Summary of the approximations for the label gain and loss rates in the POI provided here for different scenarios.** A POI with a precursor population has four qualitatively different gain and loss rate scenarios. These approximations underscore how crucial it is to know the precursor's dynamics in identifying the true turnover rate of the POI. The special cases are derived in Equations 6 and 7, while the regular cases are derived in Equations 9 and 11.

| Scenarios | $p^*(0)$ | $p^*(1)$ | $d^*(1+\varepsilon)$ | Constraints |
|---|---|---|---|---|
| **General cases** | | | | |
| **Differentiation without division** | $p_2$ | $\frac{d_2 d_1 + p_2(1-d_1)}{(1+d_2)}$ | $d_2 - (d_2 - p_2)\frac{d_1}{p^*(1)}$ | $k=1$ |
| **Division-linked differentiation** | $\frac{(d_2+p_2)}{2}$ | $\frac{d_2(d_1+1)+p_2(1-d_1)}{2(1+d_2)}$ | $d_2 - \frac{(d_2-p_2)}{2}\frac{d_1}{p^*(1)}$ | $k=2$ |
| **Special cases** | | | | |
| **Precursors turn over about as fast as deuterated water/glucose** <br> **OR** <br> **The source is negligible** | $d_2$ | $d_2$ | $d_2$ | $d_1 > d_2$ <br> $1 \ll d_1$ <br> OR <br> $d_1 \approx 0$ |
| **Rapidly turning over, non-dividing POI with precursors that do not divide while differentiating** | $d_1$ | $d_1$ | $d_1$ | $d_1 < d_2$ <br> $1 \ll d_2$ <br> $p_2 = 0$ <br> $k=1$ |

4) the average of its division and death rates, $(p_2 + d_2)/2$, if the precursors go through division-linked differentiation ($k = 2$). Here, along with the label gain due to the division in the POI, the source from the precursors also has a significant contribution of labelled cells as the precursor cells also divide and pick up label when they differentiate into the POI. Thus, if a naive T cell were to undergo division while differentiation into memory T cells [18], the gain of label by memory T cells would reflect both its division and death rates.

These four different labelling behaviours can be distinguished from each other if the turnover rates of the precursors is known. Therefore, it is essential to have information on the precursors' dynamics. Since the labelling in the POI depends on the enrichment of its immediate precursors only (equation 2h), it is sufficient to have knowledge or data on the immediate precursors (and not the precursors' precursor) to distinguish the rates of the POI from that of its precursors. So, even a phenomenological description of the precursors' labelling curve, $l_1(t)$, would suffice to correctly estimate the turnover rate of the POI. Therefore, it is essential and sufficient to measure and model the immediate precursor population of the POI to reliably interpret the estimated rates.

### Numerical tests reveal the range of variations around the identified parameter relationships that is introduced by fitting simple models to data

Up until now, we have identified relationships between the parameters of the ES model (for example, $p_2$ and $d_2$) and the parameters of the phenomenological model ($p^*$ and $d^*$), by analytically comparing equations. Next, we aimed to take the measurement noise in the labelling data and uncertainties introduced by fitting simplified models to that data into account.

We systematically varied the values of the three parameters of the ES model ($d_1$, $p_2$ and $d_2$) to generate a 1000 different parameter sets. Each parameter could have 10 different values: $d_1 = 0.5i$, $d_2 = 0.5i$, $p_2 = \alpha_2 d_2$, $\alpha_2 = 0.1i$, where $i \in \{1, \ldots, 10\}$. These 1000 parameter sets were then used to make artificial data using the ES model (having three parameters). This dense and perfect data was subsequently fitted with the implicit kinetic heterogeneity model (Equation 8, having two parameters, $p^*$ and $d^*$). As we are interested in the average turnover rate, the estimated value of $p^*$ was plotted against the true values of the three parameters of the ES model (Figs 3 and 4).

To test the analytical prediction of the label gain rate (equation 9c), we also plotted the estimated $p^*$ values against $p^*(0)$, which denotes the initial label gain rate of a population. Note that in the companion paper [10] $p^*(0)$ is referred to

as the quantity 'production by division', which combines potential division-linked differentiation of the precursors with the proliferation within the POI. Production by division is defined as $p_2 + (d_2 - p_2) \frac{(k-1)}{k}$ (i.e., the coefficients of $D(t)$ in equation 2h), which is equal to $p^*(0)$.

***If differentiation is not accompanied by cell division (i.e., k = 1).*** The relationships observed in the numerical simulations (Fig 3) tend to be in line with our analytical results. The estimated label gain rate, $p^*$, is always lower than the true turnover rate of the population of interest, $d_2$ (Fig 3b), and is approximately equal to the true proliferation rate of the population of interest, $p_2$ (Fig 3c). The initial label gain rate, $p^*(0)$, is equal to $p_2$ when $k = 1$, and hence Fig 3c and 3d are identical. The numerical results do not always match the analytical ones, as the relationship between $p^*$ and $p_2$ has a wide range for low values of $p_2$, and differs by more than 2-fold for many parameter sets (the dashed line indicates the relationships $p^* = 2p_2$ and $p^* = 2p^*(0)$ in Fig 3c and 3d, respectively). This shows that fitting a simplified model to the labelling data can introduce error in the value of the estimated parameters.

In Fig 3, the implicit kinetic heterogeneity model was fitted to a dense, artificial dataset that did not have any experimental noise. However, true deuterium labelling data are never free of noise and are not as dense as our artificial data. Reassuringly, when we used a dense dataset (Fig G in S1 Text, top row), or a sparse dataset (Fig G in S1 Text, bottom

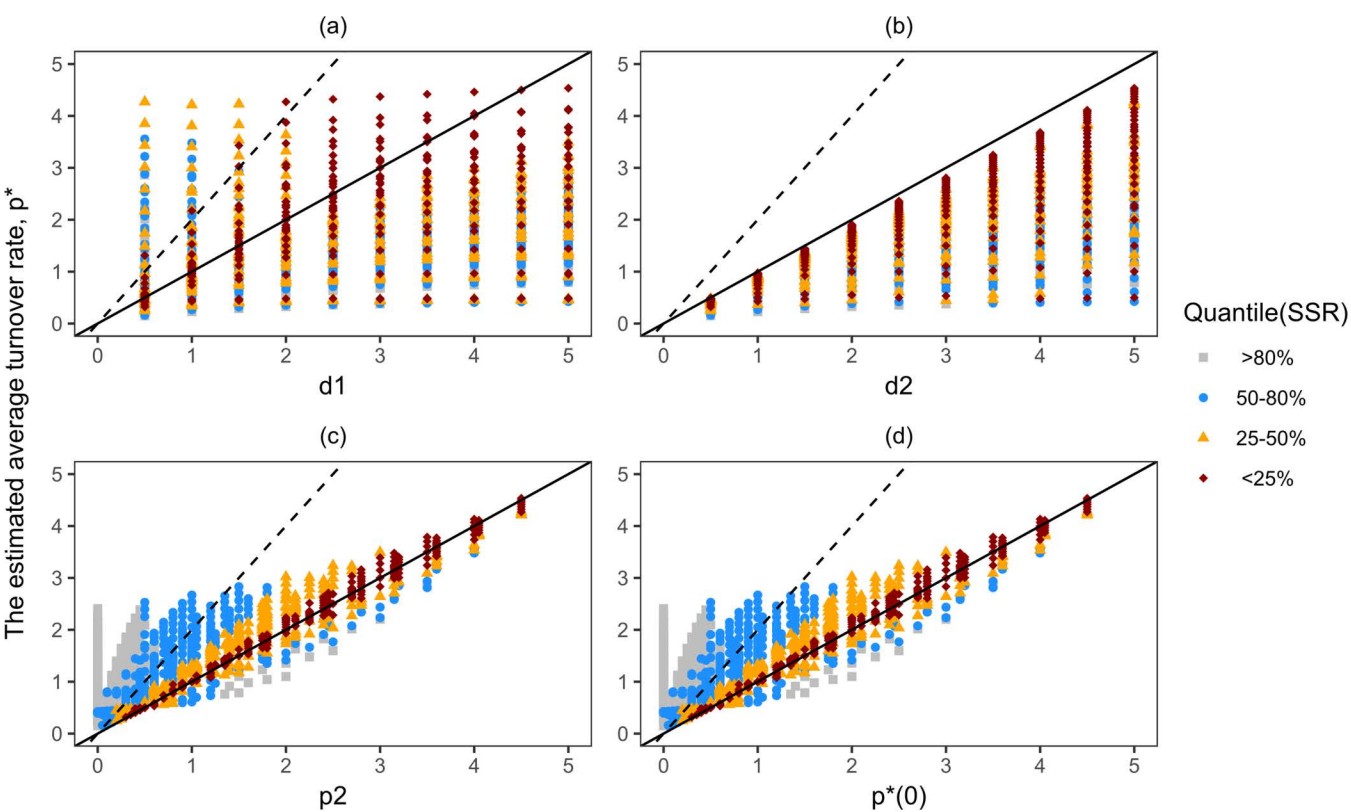

**Fig 3. The estimated turnover rate of the POI underestimates its true turnover rate and reflects its true division rate when precursors differentiate without division into the POI.** The above plots show the distribution of the estimated gain rate, $p^*$, as a function of the true rates of the precursor population and the POI when $k = 1$. The solid and dashed lines represent slopes of 1 and 2, respectively. The lines with the slope of 1 and 2 show the line where the value of $p^*$ is equal to and twice that of the parameters on the x-axis, respectively. The colours and shapes show four different groups of quantiles calculated based on the sum squared residual (SSR) of the 1000 best fits. For example, the dark red diamonds show the 0-25% quantile group, which are the top 250 best fits among the 1000 best fits.

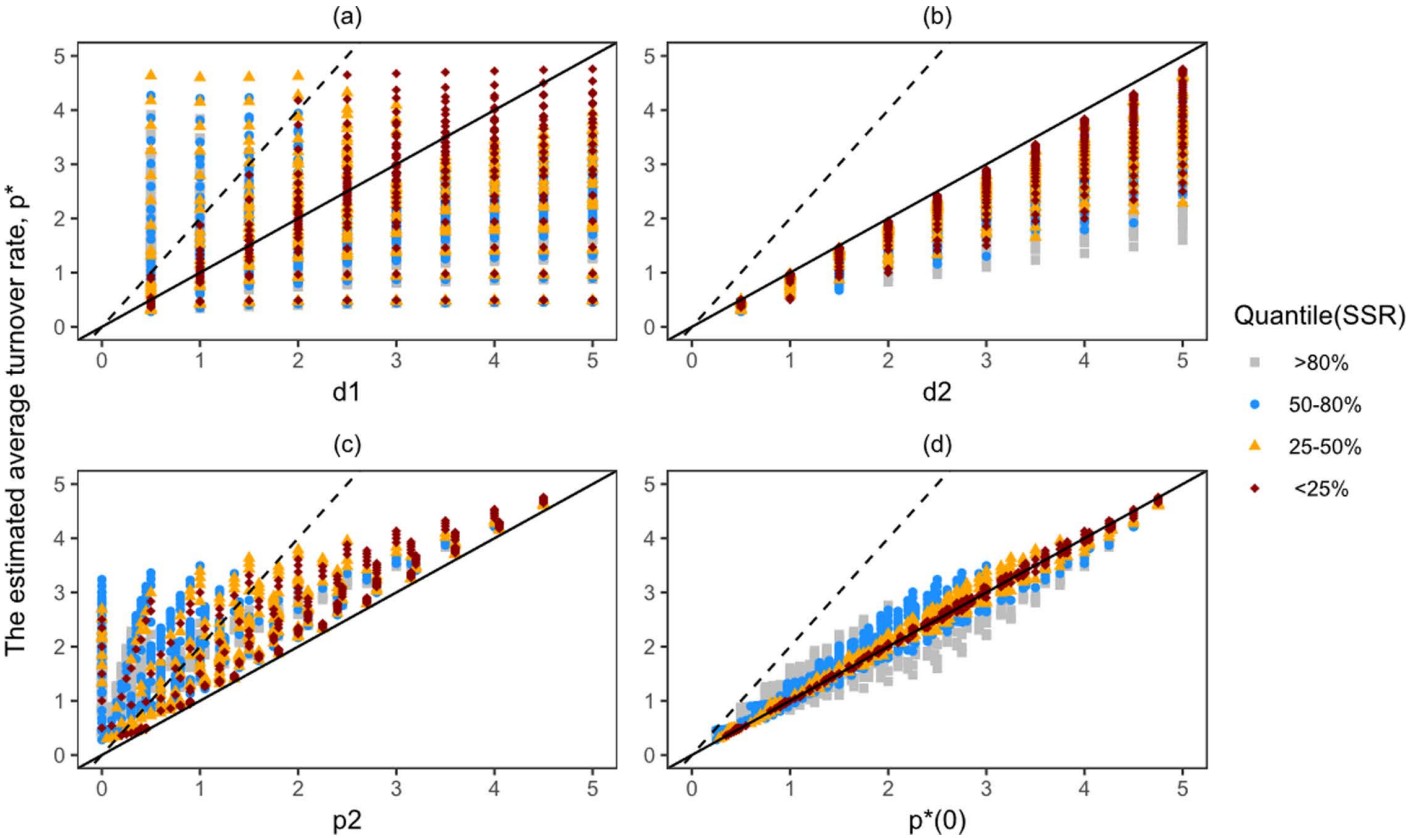

**Fig 4. The estimated turnover rate of the POI underestimates its true turnover rate and overestimates its true division rate when precursors undergo division-linked differentiation.** The above plots show the distribution of the estimated gain rate, $p^*$, as a function of the true rates of the precursor population and the POI when $k = 2$. The solid and dashed lines represent slopes of 1 and 2, respectively. The lines with the slope of 1 and 2 show the line where the value of $p^*$ is equal to and twice that of the parameters on the x-axis, respectively. The colours and shapes show four different groups of quantiles calculated based on the sum squared residual (SSR) of the 1000 best fits. For example, the dark red diamonds show the 0-25% quantile group, which are the top 250 best fits among the 1000 best fits.

row), or when we added noise to the data (Figs H and I in S1 Text), the relationships between the parameters did not change.

Since the data points in these figures above are coloured by the quantile (based on the SSR of the best fits) that those fits lay in, we can see that the deviations from the predictions are largely due to the fits in the bottom 50% quantile (see Figs G and I in S1 Text). However, as a quantile is a relative measure, this does not necessarily mean that the fits in the bottom 50% quantile poorly describe the data (see Fig J in S1 Text). Thus, this numerical exercise not only confirms the analytical results, but also identifies parameter values (for example, of $p_2$) where the estimated quantity ($p^*$) is expected to have a large error.

***If differentiation is accompanied by cell division (i.e., k = 2).*** The numerical results also tend to be in line with the analytical results when the precursors go through division-linked differentiation into the POI. The estimated label gain rate, $p^*$, always underestimates the true turnover rate of the POI, $d_2$, as in the $k = 1$ case (Fig 4b). However, unlike the $k = 1$ case, $p^*$ always overestimates the true proliferation rate of the POI, $p_2$ (Fig 4c). This was bound to happen as the estimate of $p^*$ takes both the division-linked differentiation of the precursors and the proliferation within the POI into account. The initial label gain rate, $p^*(0)$, is approximately equal to the estimated $p^*$ (Fig 4d). See Fig J in S1 Text for representative fits to dense or sparse datasets.

Qualitatively, Fig 4a–b look similar to Fig 3a–b, whereas Fig 4c–d seem to be a subset of Fig 3c–d. Thus, it appears that the $k = 2$ case is a subset of the $k = 1$ case. We will analytically confirm this below. Importantly, in both cases, the estimated $p^*$ is consistently well-captured by the parameter $p^*(0)$, meaning that we can always estimate $p^*(0)$ from deuterium labelling data using the simple implicit kinetic heterogeneity model. However, the true turnover rate of a POI, $d_2$, cannot be inferred from $p^*(0)$ as the value of $p^*(0)$ depends on both $p_2$ and $d_2$, which weakens the significance of estimating this identifiable parameter.

---

### Box 1. Danger zone of deuterium labelling interpretation

The incorporation of deuterium in the POI is dependent on various factors: the rate at which the precursors are replaced, whether precursors undergo cell division when differentiating into the POI, the rate of replacement of cells in the POI, the rate at which cells in the POI divide, and the length of the labelling period. The most important comparison among these factors that define the interpretation of the deuterium labelling data are the replacement rates (also referred to as turnover rates, and defined as the reciprocal of the expected lifespan) of the POI and the precursors. To better understand this, let us consider the following cases (see Fig 5 below):

1. Short-lived precursors: If the precursors are so fast that the labelling levels of the precursors reflect the body deuterium levels, the rate at which the POI gains label reflects the true turnover of the POI. As an example, the lifespan estimates of naive T cells in the literature are likely to be accurate as it is well-known that short-lived thymocytes are the precursors of naive T cells. The point ($d_2 \approx 0.025 = 2/80$, $d_1 \approx 8$) is annotated with an 'N' as the estimated lifespan of naive T cells in humans is ~80 months (6.5 years) and that of thymocytes is in the order of days based on a 2-months long deuterium labelling experiment [14].

2. Short-lived POI: If the cells in the POI are replaced more quickly than their precursors, the estimated turnover rate of the POI reflects the turnover rate of its precursors (provided that cells in the POI are not dividing). The case of neutrophils is an ideal example here (annotated at ($d_2 \approx 0.5 = 6/12$, $d_1 \approx 0.07 = 6/84$) as 'G' for granulocytes as the most widely accepted estimated lifespan of blood neutrophils in humans is ~12 hours and that of neutrophil precursors is ~0.5 week (84 hours) based on a 6-hour deuterium labelling experiment [4]). The rate at which deuterium is lost from blood neutrophils may either reflect the rate at which blood neutrophils are replaced or the rate at which their precursors in the bone marrow turn over [4,9]. The current dogma dictates that the precursors in the bone marrow have a slower turnover rate than blood neutrophils. However, in the absence of any information on the dynamics of the precursors in bone marrow, it remains impossible to estimate the lifespan of blood neutrophils.

3. All other situations: If none of the populations (i.e., the precursor or the POI) is much short-lived than the other, the estimated turnover rate reflects a combination of the POI's and precursor's rates (see Table 2 in the manuscript for the approximations). It is likely that most cell populations fall in this category. For example, deuterium labelling curves of memory T cells (annotated at ($d_2 \approx 0.3 = 2/6$, $d_1 \approx 0.025 = 2/80$) as 'M' as the estimated lifespan of memory T cells in humans is ~6 months, with naive T cells as their most likely precursors, based on a 2-months long deuterium labelling experiment [12]) typically show that a considerable fraction of cells remains unlabelled even after four weeks of labelling. Furthermore, there is still no consensus on the precursors of memory T cells in different subpopulations, making it difficult to interpret the turnover rates estimated from the deuterium labelling curves of memory T cells. The (current) best known precursors of memory T cells are naive T cells, which replace a small portion of the memory T-cell pool every month, both in mice [18] and in humans (in preparation)

The colours (in Fig 5) represent how wrong the estimated turnover rate could be if the turnover rate of the precursors would not be taken into account. Unfortunately, the danger zone is much larger than the safe zone and most studied cell

populations lie in the danger zone. Our results show that misinterpreting the rate at which the POI gains label as its turnover rate could lead to a drastic (i.e., up to 7-fold) overestimate of the POI's lifespan (see Fig 6). It is, therefore, important to carefully re-examine existing lifespan estimates of different cell populations by taking the dynamics of their precursors into account.

### The peaks of the labelling curves can help distinguish scenarios

**Timing of the peak.** Interestingly, it is possible to infer some properties of the precursors and the POI just by comparing both the height and the time of their peaks. The POI reaches its peak enrichment either precisely at, or sometime after, the

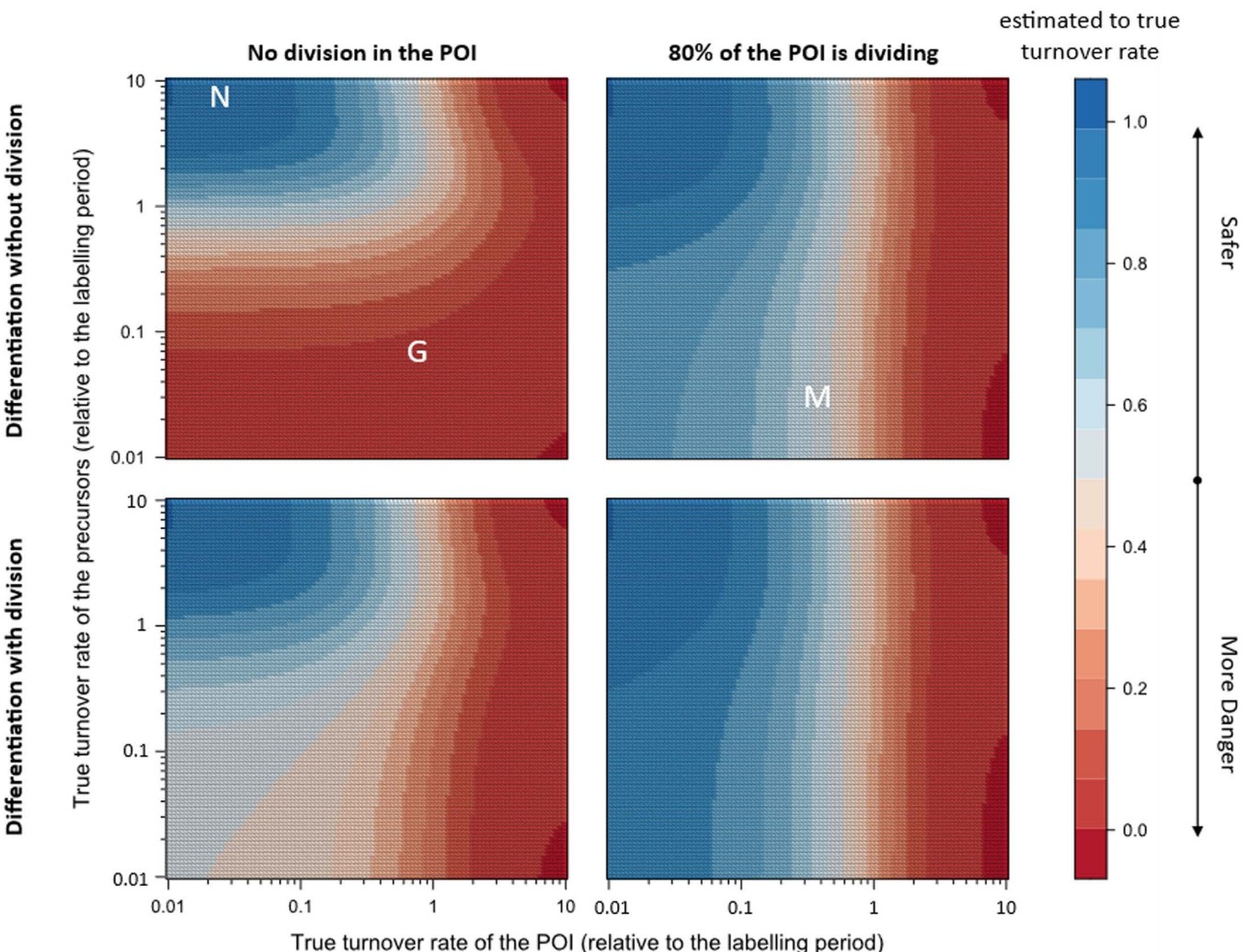

**Fig 5. The danger and safe regions in the space defined by the turnover rates of the POI and the precursor populations, along with annotations (in white) of a few well-known cell populations (N: naive T cells, G: granulocytes/neutrophils, M: memory T cells).** This heatmap was generated using the approximation of the label gain rate at the end of the labelling period as derived in this manuscript (i.e., $p^*(t)$ with $t = 1$ from equation 9b). The deuterium labelling curve of any cell population outside the dark blue region must be analyzed along with that of its precursors to accurately estimate the POI's rates.

end of the labelling period. For a model with random cellular processes (i.e., an ODE-based model), the peak of the POI will be *at the end* of the labelling phase when the precursors flowing into the POI have lower enrichment than the cells in the POI, i.e., $l_2(1) > l_1(1) \Leftrightarrow p^*(0) > d_1$ (from Equation 10). If precursors differentiate without dividing (i.e., when $k = 1$),

$$p^*(0) > d_1 \Leftrightarrow p_2 > d_1 \tag{13}$$

i.e., the peak will be at the end of the labelling phase when the division rate of the cells in the POI is higher than the loss rate of the precursors. Instead, if the differentiation of the precursors is accompanied by division (i.e., when $k = 2$),

$$p^*(0) > d_1 \Rightarrow \begin{cases} p_2 > d_1, \text{ or} \\ d_2/2 > d_1 > p_2 \end{cases} \tag{14}$$

meaning that the peak will be at the end of the labelling phase when the division rate of the POI is either higher than the loss rate of the precursors, $p_2 > d_1$, or is lower than the loss rate of the precursors, and the loss rate of the precursors is lower than half of the loss rate of the POI, $d_2/2 > d_1 > p_2$. Summarizing, the peak of the POI is at the end of the labelling period only when the enrichment of the POI is largely due to division (within the POI or by differentiating precursors).

There are two corollaries of the above equations (Equations 13 and 14). One, a non-dividing POI always achieves its peak labelling after the stop of labelling. Two, the peak of the POI will be *after the end* of the labelling phase if the precursor cells flowing into the POI at and after the end of the labelling phase have a higher enrichment than the cells in the POI, i.e.,

$$l_1(1) > l_2(1) \Rightarrow d_1 > p^*(0)$$

$$\Rightarrow \begin{cases} d_1 > d_2, \text{ or} \\ d_2 > d_1 > p_2 \end{cases} \quad \text{when } k = 1 \tag{15}$$

$$\Rightarrow \begin{cases} d_1 > d_2, \text{ or} \\ 2d_1 > d_2 > d_1 > p_2 \end{cases} \quad \text{when } k = 2$$

Thus, the peak of the POI is after the end of the labelling phase only when non-dividing precursors differentiating into the POI are the major contributor of label in the POI. Therefore, the population architecture underlying a labelling curve can be partially realized by comparing the location of the peaks of the POI and its precursors (provided the dataset is sufficiently dense).

**Division-linked differentiation.** To further pinpoint the properties of the precursors and the POI, it is important to be able to distinguish between labelling curves where differentiation is accompanied by division from those where it is not. Unfortunately, whether a population goes through division-linked differentiation is, in general, unidentifiable (i.e., curves with $k = 1$ and $k = 2$ are indistinguishable). To demonstrate this, we compare the labelling equations of the POI when $k = 1$ and $k = 2$ (equation 2h). Denoting the division rate of the POI, $p_2$, as $p_{21}$ when $k = 1$, and as $p_{22}$ when $k = 2$, we find

$$\left.\frac{dl_2(t)}{dt}\right|_{k=1} = \left.\frac{dl_2(t)}{dt}\right|_{k=2} \tag{16a}$$

$$\Rightarrow d_2\left(l_1 - l_2\right) + p_{21}(D(t) - l_1) = \frac{d_2\left(l_1 + D(t) - 2l_2\right) + p_{22}(D(t) - l_1)}{2} \tag{16b}$$

$$\Rightarrow p_{22} = 2p_{21} - d_2 \tag{16c}$$

where $p_{21} \geq d_2/2$. Therefore, the labelling curve of any population whose precursor goes through division-linked differentiation can also be explained by a scenario where the precursors differentiate without division (see Table 1 for examples).

On the other hand, the labelling curve of a population whose precursors do not go through division-linked differentiation cannot be described with $k = 2$ if more than half of the production of the POI is due to the source, i.e., if $p_2 < d_2/2$. Thus, one can conclude that the influx into the POI is not accompanied by division if $p_2 < d_2/2$ in the estimates of the best fit. In a scenario where it is not known whether differentiation is linked with division, it is safe to conclude that the POI is primarily maintained by self-proliferation only if $p_2 \geq 3d_2/4$. Fortunately, the above transformation (Equation 16) does not affect the turnover rate of the POI, $d_2$. Therefore, the turnover rate of the POI is identifiable if (and only if) the labelling in the precursors is known.

**Intersecting labelling curves.** Consider a simple scenario where precursors differentiate without division into a non-dividing POI, i.e., $k = 1$ and $p_2 = 0$ (see above). We have seen above that the peak enrichment in the POI will then be reached after the end of the labelling phase (where $D(t > 1) = 0$). More importantly, the peak occurs when $I_2(t) = I_1(t)$ (Equation 4). Conversely, if $I_2(t) = I_1(t)$ when $t > 1$, equation 2i boils down to

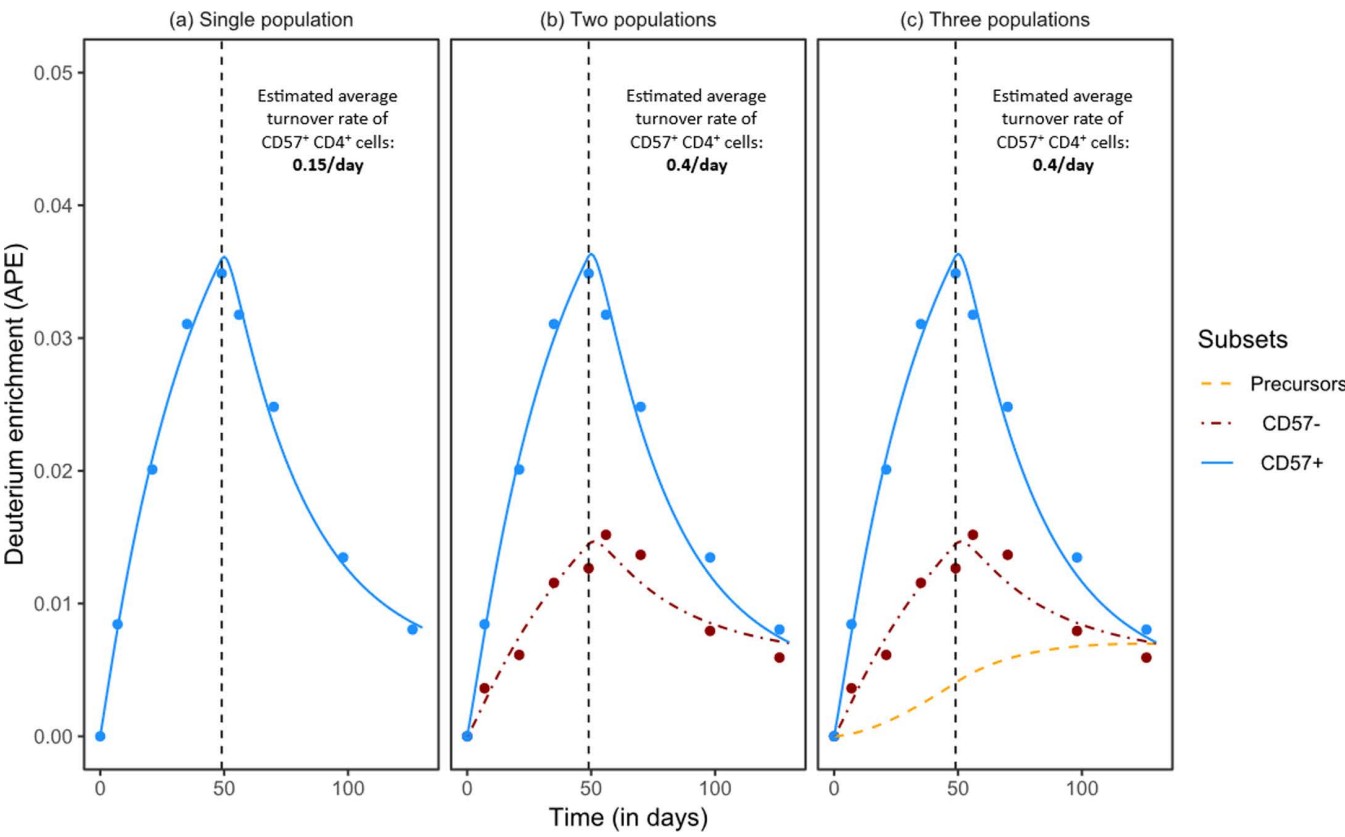

**Fig 6. Similar description of the deuterium labelled fraction of CD57$^+$CD4$^+$ T cells based on different models despite radically different estimated rates.** The above plots show the best fits of the two sub-population kinetic heterogeneity model (a), the two sub-population kinetic heterogeneity and the one population ES model (b), and the two sub-population kinetic heterogeneity and the two population ES model (c) to the deuterated-water labelling data of CD57$^-$CD4$^+$ and CD57$^+$CD4$^+$ memory T cells of individual DW02 from Ahmed et al. (2020) [15]. See S1 Text for the system of equations (equation E). The labelling data of the cell populations and the data of the body deuterium concentration were digitized from the original article for re-analysis. The parameters describing the deuterium concentration in the body water of individual DW02 are $f = 0.031$, $\delta = 0.032$/day, $\beta = 0.0023$, where $f$ is the predicted asymptote of deuterium enrichment in body water, $\delta$ is the estimated turnover rate of water and $\beta$ is the estimated initial deuterium concentration in body water. See [15] for the equations. The above fits were based on the assumption that cells differentiate from one population into the other without division, i.e., $k = 1$. Note that the vertical axis gives the non-normalized deuterium enrichment (APE; atom percent excess) measured in the population, as was reported in the original work [15].

$$k = 1 - \frac{p_2}{d_2}$$

(17)

As $k$ takes only integer values and $p_2 < d_2$, the only value that satisfies the identity in Equation 17 is $p_2 = 0$, implying $k = 1$. This provides a cardinal property of a chain of populations:

$$l_2(t) = l_1(t) \Leftrightarrow k = 1, \ p_2 = 0$$

(18)

i.e., if during the de-labelling phase, a population reaches its peak when its labelling curve intersects that of its precursors, then this is proof that the POI is not dividing, and that its precursors differentiate into the population without division. This immediately implies that such a population ($k = 1$, $p_2 = 0$) has a lower peak than its precursor.

In certain cases, it is possible to infer the model architecture from the labelling curves. For example, it is safe to conclude that the cells in a population are not governed by random processes if the peak of the population is higher than its precursor and the peak is reached after the end of the labelling period (see Fig K in S1 Text, equations C-D and the associated text in S1 Text).

### Case study confirms that the underlying maintenance mechanisms can have enormous effect on the estimated turnover rate of the POI

The ES model can explain a wide variety of labelling curves, including the curves generated by the IS model. Both $p^* \leq d^*$ and $p^* > d^*$ labelling behaviour can be explained by the ES model (Table 2). This generality of the ES model is troubling, however. Most current estimates of cell lifespans are based on IS-like models that do not consider a precursor population, and the estimated rate at which label is gained is generally interpreted as the average turnover rate of the POI [5,6]. Since in the ES model, the same estimate could be reflecting the rate of the precursors ($d_1$), of the POI ($d_2$, $p_2$), or anything in between, this questions some of the current interpretations. We illustrate this problem with an example.

### Labelling of CD57+ memory T cells by Ahmed at al. (2020)

A dataset that seems uniquely suited for our analysis was generated by Ahmed et al. (2020) who measured the label gain and loss in both the POI, i.e., CD57+CD4+ memory T cells, and their precursor population, i.e., CD57-CD4+ memory T cells [15]. To find out the dynamics of the POI, they used a phenomenological chain of two populations describing both the POI and the precursors (see [15] for their model). Their analysis showed that both the precursors and the POI were replaced rapidly (on average every 2 days) and that the POI was maintained largely (~ 95%) by division rather than by the source from the precursors (~5%). Hence, the maintenance of the POI and its precursors were considered to be independent.

To re-analyse these data, we considered three different scenarios (Fig 6): modelling only the POI (Fig 6a); the POI and its precursors (Fig 6b); and the POI, its precursors and the precursor's precursor (Fig 6c). The first population in each of these scenarios (i.e., CD57+CD4+ in Fig 6a, CD57-CD4+ in Fig 6b and the precursors of CD57-CD4+ in Fig 6c) is always described with the two sub-population kinetic heterogeneity model (Equation 3). The other populations (i.e., CD57+CD4+ in Fig 6b, and both CD57-CD4+ and CD57+CD4+ cells in Fig 6c) were modelled with the ES model (equation 2h).

The labelling of CD57+CD4+ cells was well-described in all cases. However, the estimated rates were very different depending on the model used to describe the data and whether its precursors were explicitly modelled. The CD57+CD4+ population was estimated to have a 6.67-day lifespan when the population was described as a two sub-population kinetically heterogeneous population, which was ~3-fold longer than when the POI was described as a homogeneous model with a partially labelled precursor population (Fig 6a versus Fig 6b and 6c, Table 3). Similarly, the estimated lifespan of CD57-CD4+ cells was 7-fold longer if the model was not given the freedom to opt for a partially labelled precursor population (Fig 6b versus 6c, Table 3). Note that the description and the estimates of the CD57+CD4+ cells were identical for

Table 3. Parameter estimates of the best fits shown in Fig 6. The quantities $\delta_1$, $\delta_2$ and $\alpha$ are the parameters of the two sub-population kinetic heterogeneity model (Equation 3). The average turnover rate is denoted by $\bar{\delta}$ and is calculated as $\alpha\delta_1 + (1-\alpha)\delta_2$. The quantities $d_2$, $p_2$, $d_3$ and $p_3$ are the parameters of the ES model (equation 2h). The rates are reported in per day units. The AICs of the single population (a), two populations (b), and three populations (c) fits are -78.80, -261.47, and -257.47, respectively. The quantity of interest, i.e., the estimated turnover rate of CD57⁺CD4⁺ memory T cells, for the three scenarios is shown in bold.

| | (a)Single population | (b)Two populations | (c)Three populations |
|---|---|---|---|
| **Precursors** | – | – | $\bar{\delta} = 0.001$ <br> $(\delta_1 = 0.001, \delta_2 = 1, \alpha = 1)$ |
| **CD57⁻CD4⁺** | – | $\bar{\delta} = 0.04$ <br> $(\delta_1 = 0.28, \delta_2 = 0.001, \alpha = 0.13)$ | $d_2 = 0.28,$ <br> $p_2 = 0.04$ |
| **CD57⁺CD4⁺** | $\bar{\delta} = \mathbf{0.15}$ <br> $(\delta_1 = 0.37, \delta_2 = 0.002, \alpha = 0.39)$ | $\mathbf{d_2 = 0.4,}$ <br> $p_2 = 0.12$ | $\mathbf{d_3 = 0.4,}$ <br> $p_3 = 0.12$ |

the two cases in which the population was described with the ES model. This is because the description of the label in precursors (CD57⁻CD4⁺ cells) was identical, even though the estimated rates for the precursors were different (Fig 6b and 6c). Thus, only knowing the labelling dynamics of the precursors (without knowing its true turnover rate) is enough to find a reliable estimate of the POI's true turnover rate.

One of the primary takeaways of the original paper was that most of the CD57⁺CD4⁺ population was replaced by division. In contrast, the estimates found from the fit of the ES model to the labelling data of CD57⁺CD4⁺ cells suggest that only 31% of the population is replaced due to division ($p_2/d_2$ in Fig 6b and $p_3/d_3$ in Fig 6c), which is very different from the original estimate of 93% (for individual DW02), illustrating that mechanistically allowing for precursors can radically affect the biological interpretation of the data.

The precursors (and the precursor's precursor) in the above fits were considered to differentiate without division into the POI (and into the precursors), i.e., $k = 1$. Since both the CD57⁻CD4⁺ (~86%) and the CD57⁺CD4⁺ (~69%) memory T-cell populations were estimated to be largely maintained by the source (Table 3), our analytical derivation in the section ***Division-linked differentiation*** implies that this data cannot be described well with a model where differentiation of precursors is accompanied by cell division. A poor description of the data by the model with $k = 2$ confirms our analytical results (Fig L in S1 Text). In contrast to the original article, which suggested a clonal burst during the differentiation of CD57⁻CD4⁺ cells into CD57⁺CD4⁺ cells, the ES model suggests that precursors merely differentiate (i.e., without division) into the POI. Therefore, this case study also provides an example where only the deuterium labelling data is enough to deduce whether a population undergoes division-linked differentiation.

The above example nicely illustrates that the estimated parameters can strongly depend on the underlying model, and hence that one should always test whether the estimates change when a precursor with a realistic turnover rate is added. The estimates found from the ES model, in this example, could be realistic as they are in line with previous estimates stating that as much as a quarter of the CD4⁺ memory T-cell population could be maintained by a source from naive T cells [18].

## Discussion

Deuterium labelling is hailed as the current best technique to accurately measure the turnover rates of a cell population *in vivo* [4]. In this article, we share some insights into what can and what cannot be deduced from deuterium labelling data using the current modelling approaches. We show that models that fail to consider a slow source of label can severely underestimate the true turnover rate of the POI. The rate at which the POI gains label is not simply its average turnover rate but can be markedly influenced by the rate at which its immediate precursor population gains label. We also show that typically one cannot tell from the labelling data whether cells divide while differentiating into the POI. Therefore, unless one is convinced that 1) the precursors are much faster, or 2) hardly play any role in maintaining the POI, a source should be modelled explicitly (with the ES model) and ideally be matched with the labelling data of the precursors.

Our analyses show that the rate at which the POI gains label is always lower or equal to the true turnover rate of the POI. Depending on the dynamics of its immediate precursor population, the labelling in the POI can follow 4 different scenarios. In cases where the lifespans of both the precursors and the POI are comparable to the labelling duration, the rate of labelling varies with time and is not given by the rates of any single population. So, we defined two label gain rates, one at the beginning of the labelling period, $p^*(0)$, and the other at its end, $p^*(1)$. We show that these approximated gain rates tend to be in good agreement with each other, and that both are smaller than the turnover rate, $d_2$, of the POI. Thus, interpreting the label gain rate as the turnover rate can markedly overestimate the lifespan of the measured population.

There are two cases where one can be certain that the estimated rate reflects a POI's true turnover rate. First, when the source population's turnover rate is sufficiently fast, the labelling in the POI is largely dictated by its own turnover rate. For example, thymocytes, that act as a precursor to slowly turning-over naive T cells, turn over rapidly. Therefore, estimating the true turnover rate of naive T cells does not require measuring the labelling in the thymocytes. Second, when the population is largely maintained by proliferation (i.e., when the source into the POI is negligible). Memory T cells are a likely example of this scenario as they are thought to be largely maintained by proliferation. Therefore, although measuring the label incorporation in the precursors should be treated as the standard procedure, it may not be necessary in all deuterium labelling experiments. Note that there's also the case where precursor cells clonally expand during their differentiation into the POI cells (see the companion paper, *Yan et al.* [10]). As all the cells flowing into the POI, in this case, would be labelled, this scenario would approximate the first case mentioned above where the source is highly labelled.

In this manuscript, we presented analyses based on calculated slopes at the beginning and end of the labelling period. This was important to find general conditions that distinguish deuterium labelling curves and to predict the underlying mechanisms. When analyzing real-world deuterium labelling experiments, one can just fit the ES model to the data. Using a study on the maintenance mechanism of CD4$^+$CD57$^+$ memory T cells [15] as a case study, we showed how estimates can become very different when the source population is not extremely rapid. The ES model suggested that these cells were only partly (~31%) maintained by division, as opposed to the original interpretation that suggested that they were largely maintained by self-replication. This is similar to previous conclusions on CD4$^+$ memory T cell pool, that are maintained by an influx from the naive CD4$^+$ T cell pool [18]. Additionally, in contrast to the original conclusion where the differentiation of CD4$^+$CD57$^-$ cells was accompanied by a clonal burst, the ES model showed that the labelling of both CD4$^+$CD57$^-$ cells and CD4$^+$CD57$^+$ cells are best-described if the differentiation of CD4$^+$CD57$^-$ cells is not accompanied by division. In this case study, we found that the estimated lifespans could differ by as much as 7-fold if the precursors were allowed to be partially labelled, underlying the importance of considering an explicitly modelled source when analysing deuterium labelling data.

In an independent study in the same issue as this article, *Yan et al.* [10] also argue in favour of explicitly modelling the precursor population of the POI. Whereas we allow for maximally one cell division upon differentiation of a precursor cell into the POI (i.e., $k = 1$ or $k = 2$), *Yan et al.* [10] also consider precursors that go through extensive clonal expansion ($k = 2^i$, where $i \geq 0$ in their paper, meaning that the expansion is instantaneous). Interestingly, regardless of this difference, our results are similar. This could be because the more general $k = 1$ case includes the $k > 1$ cases. We showed, here, that the case of $k = 1$ includes the $k = 2$ case. It would be interesting to see whether it also includes the $k > 2$ cases. Yan and colleagues find the same relationship between the estimated turnover rate of the POI and its true proliferation rate as we do, i.e., $p^* = p_2$ (when $k = 1$) and $p^* > p_2$ (when $k = 2$). They also conclude that the turnover rate of the POI, $d_2$, is not equal to the estimate of $p^*$, however, unlike us they only conclude that for the $k = 1$ case. They show that for $k > 1$, the turnover rate of the POI is well-approximated by the estimated value of $p^*$, i.e., $p^* \approx d_2$. This is likely due to the difference in our assumptions about maximally one division (in our case) and clonal expansion [10], as most of their data samples could reflect cases where $k > 2$. If $k$ is high, the enrichment in the precursor cells flowing into the POI would be very high (close to the maximum), essentially boiling down the ES model into the IS model, in which case it would be expected that $p^* \approx d_2$. The agreement between the conclusions drawn by both studies present a compelling case for explicitly modelling the precursor population of a POI that is at homeostasis.

Our work has a several limitations. First, the linear approximations are only valid for the non-saturated labelling curves, implying that the turnover of the precursors and the POI should be sufficiently slow compared to the labelling period. Fortunately, this is true for most well-designed labelling experiments. Second, the approximations are accurate only if the deuterium enrichment in plasma is fast enough to be approximated well by a step function. Third, our proposed solution to take into account the turnover of the precursors may not always be feasible, since the true precursors of a POI may not be known. In fact, our results here show that the estimate of a POI's lifespan can be very sensitive to the turnover of its precursor population and may, thus, have to be revised when new insights about the true precursors of a POI have been gained. We do not discuss ways of finding the best precursor candidate here, as this is out of the scope of the current study. Instead, our work provides a framework to calculate the expected variation in the lifespan of a POI due to its precursors, again underlining why one should never ignore the precursors.

The true dynamics of the deuterium concentration in the body are dependent on the source of the label: deuterated glucose or deuterated water. In experiments with deuterated glucose, the body deuterium concentration is described very well by a step function (as considered in this manuscript) because glucose turns over rapidly [8,19]. The step function description of the body deuterium concentration is less accurate for experiments that use deuterated water, because the turnover rate of water is much slower than that of glucose (4-day timescale in mice and 10-day timescale in humans) [6,12,19]. The relatively slow loss of deuterium from the body water results in cell populations attaining their peak label intensity after the labelling period has ended. In such cases, the approximation for the initial label loss rate can become less accurate. Fortunately, even if the true dynamics of the body deuterium concentration are taken into account, the approximations for the label gain rate of the POI (both at the beginning as well as at the end of the labelling period) presented here remain valid (Figs B-D in S1 Text).

Our results emphasize the need to always take into account the dynamics of the precursor population when interpreting the dynamics of the POI. Unfortunately, the precursor population of a POI is not always known, and even for cell populations for which we think we know the precursor, insights may change, and hence require reinterpretation of the dynamics of the POI. In fact, deuterium labelling data can even be used to help identify the most likely precursor population of a POI. Further, whether differentiation of precursors into the POI is accompanied with division is seldom known. We have shown that inaccuracies in any of these quantities can lead to estimates that are as much as 7-fold off, underscoring the importance of identifying (and knowing the dynamics of) the true precursors of a POI. Importantly, our work suggests that even taking into account the (expected) labelling of the best precursor candidates of a POI is better than just fitting the IS model that assumes fully-labelled precursors. Future studies into cellular dynamics should, therefore, never neglect the dynamics of the current best-known precursors of the POI.

Taken together, we here we show that even the interpretation of deuterium labelling data, the current state-of-the-art method to estimate cellular lifespans, can be prone to error. For several different scenarios, it is essential to have information on the dynamics of the precursors to not undermine the reliability of the estimates of the POI based on deuterium labelling experiments. Fortunately, we found that it is sufficient to know the fraction of labelled cells (and not necessarily the true turnover) of the immediate precursor population among a possibly long chain of precursor populations to identify the POI's true *per capita* rates.

## Supporting information

**S1 Text.   Fig A: The estimated labelling rates when the POI is very short-lived.** The loss rate of the POI was chosen to be 20 times higher than that of the precursors ($d_1 = 0.5$, $d_2 = 10$). Note that the slopes are calculated as the label (gain or) loss rate multiplied by the fraction of (un)labelled cells, for example, $d^*(1+ \in)l_2$. The division rate of the POI, $p_2$, was either set to 5 (in (a) and (c)) or 0 (in (b) and (d)). **Fig B: Calculated gain rates are a good approximation of the true labelling curves if precursors are faster than the POI.** The turnover of deuterium in body water has a 10-day timescale in these simulations. See Fig 1 in the main text for other details. Note that the body water deuterium concentration, $D(t)$,

has been plotted on a different axis (on the right) for easier comparison to Fig 1 in the main text. **Fig C: Calculated gain rates are a good approximation of the true labelling curves if precursors are slower than the POI.** The turnover of deuterium in body water has a 10-day timescale in these simulations. See Fig 2 in the main text for other details. Note that the body water deuterium concentration, $D(t)$, has been plotted on a different axis (on the right) for easier comparison to Fig 2 in the main text. **Fig D: The estimated labelling rates if the POI is very short-lived.** The turnover of deuterium in body water has a 10-day timescale in these simulations. See Fig A in S1 Text above for other details. Note that the body water deuterium concentration, $D(t)$, has been plotted on a different axis (on the right) for easier comparison to Fig A in S1 Text above. **Fig E: The best fits of the phenomenological model (Equation 8 in the main text) to the labelling curve of the POIs shown in** Fig 1. The black circles show the data, and the red trajectory shows the best fit. **Fig F: The best fits of the phenomenological model (Equation 8 in the main text) to the labelling curve of the POIs shown in** Fig 2. The black circles show the data, and the red trajectory shows the best fit. **Fig G: Examples of the best fits used to generate** Fig 3. The top row shows fits to dense data and the bottom row shows fits to sparse data. Note that the fits and the estimates hardly change if the data is made sparser. **Fig H: Recreation of** Fig 3 **of the main text with 'noisy' data points.** The same as Fig 3 in the main text, except that each data point in the artificially generated dataset is replaced by two 'noisy' data points. The two noisy data points are drawn from a normal distribution with a mean that equals the original data point and a 20% standard deviation. The solid and dashed lines represent slopes of 1 and 2, respectively. **Fig I: Examples of the best fits used to generate Fig H in S1 Text.** The top row shows fits to dense data and the bottom row shows fits to sparse data. Note that the fits and the estimates hardly change if the data is made more sparse. The open red circles show the data points without noise added to them, while the black bullet points are the data points after noise was introduced. There are two black bullet points corresponding to each open red circle. The model was fitted to the black bullet points. **Fig J: Examples of the best fits used to generate** Fig 4. The top row shows fits to dense data and the bottom row shows fits to sparse data. Note that the fits and the estimates hardly change if the data is made more sparse. **Fig K: Non-dividing populations with the same differentiation properties show different behaviours if they have pre-programmed cellular processes.** A 3-population system is shown where cells of either all populations ((a), equation C) or only the first population ((b), equation D) have random cellular processes. The parameters of the models are: $d_1 = 0.1, \frac{1}{d_2} = \Delta_2 = 5, \frac{1}{d_3} = \Delta_3 = 4$. The values are scaled with respect to the labelling period. **Fig L: Best fits showing poor description of the labelling data of CD4 + CD57 + memory T cells if k = 2.** See captions of Fig 6 in the main text for additional details. **Table A: The true and calculated rates corresponding to the simulations shown in Fig A in S1 Text.** The rates (expressed as stu) are scaled with respect to the labelling period.
(PDF)

## Acknowledgments

We are very grateful to Ada Yan, Becca Asquith and Christiaan H. van Dorp for helpful discussions.

## Author contributions

**Conceptualization:** Arpit C. Swain, José A. M. Borghans, Rob J. de Boer.

**Data curation:** Arpit C. Swain.

**Formal analysis:** Arpit C. Swain.

**Funding acquisition:** José A. M. Borghans, Rob J. de Boer.

**Investigation:** Arpit C. Swain.

**Methodology:** Arpit C. Swain, José A. M. Borghans, Rob J. de Boer.

**Project administration:** José A. M. Borghans, Rob J. de Boer.

**Resources:** José A. M. Borghans, Rob J. de Boer.

**Software:** Arpit C. Swain.

**Supervision:** José A. M. Borghans, Rob J. de Boer.

**Validation:** Arpit C. Swain.

**Visualization:** Arpit C. Swain.

**Writing – original draft:** Arpit C. Swain.

**Writing – review & editing:** Arpit C. Swain, José A. M. Borghans, Rob J. de Boer.

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
