## [Decision Letter · Decision Letter 0]

22 Jul 2024

Dear Dr. Swain,

Thank you very much for submitting your manuscript "How are estimated cellular turnover rates influenced by the dynamics of a source population?" for consideration at PLOS Computational Biology.

As with all papers reviewed by the journal, your manuscript was reviewed by members of the editorial board and by several independent reviewers. In light of the reviews (below this email), we would like to invite the resubmission of a significantly-revised version that takes into account the reviewers' pertinent comments and suggestions. Overall, the feedback is positive, but addressing some of the issues is very important. In particular, please emphasize novelty of work versus your previous work and the unpublished Yan et al paper mentioned by multiple reviewers.

We cannot make any decision about publication until we have seen the revised manuscript and your response to the reviewers' comments. Your revised manuscript is also likely to be sent to reviewers for further evaluation.

Sincerely,

Ruy M. Ribeiro

Academic Editor

PLOS Computational Biology

Amber Smith

Section Editor

PLOS Computational Biology

Your manuscript has been carefully reviewed by three independent scientists, who made some pertinent comments and suggestions. Overall, the feedback is positive, but addressing some of the issues is very important. In particular, please emphasize novelty of work versus your previous work and the unpublished Yan et al paper mentioned by multiple reviewers.

Reviewer's Responses to Questions

**Comments to the Authors:**

Reviewer #1: Positive feedback. The paper is well written for the most parts and I could follow mathematical arguments of the authors (although I must admit I did not repeat their derivations). Figures are ok readable (captions need to be improved) and combination of analytical results with simulations is nice. The overall topic is somewhat reasonable (although for those of us studying lymphocyte turnover conclusions are pretty obvious). An example of how one could experimentally discriminate between alternative models (peak of labeling) is a nice example of good thinking, and I encourage authors to do more of that.

Major comments

Specifics of modeling the source/precursors. The authors assumed that there is only one set of precursors before POI. This is probably biologically unrealistic for immune cells (e.g., lymphocytes, neutrophils, etc) since there are many steps where cells move from less differentiated state to more differentiated state. How incorporating n precursor populations, rather than 1, changes the modeling predictions? Please explore.

Impact of deuterium dynamics on results. All results are generated assuming that labeling is described by the step function. For example, the prediction that one could discriminate between alternative models is due to peak of labeling curves is interesting but I am not sure if it would survive given complexity of deuterium accumulation over time in the system/blood. Please investigate if this complexity would negate ability to discriminate between alternatives.

Differentiation pathway. One of the main conclusions of the paper is that we need to know differentiation pathway of cells to the POI (i.e., labeling of precursors) in order to make proper inferences. I wonder if we ever know these pathways. Some recent papers challenge classical ways of hematopoietic cell differentiation (PMID: 24508463). So my question is how precisely do we need to know differentiation pathway? Or how many imprecisions in the knowledge of the pathway could there be so that turnover rate of POI is estimated with small or large errors? Some insights (discussion?) of this is needed if authors to make the conclusion they make.

Comparing models to data. It is nice that authors compare models to data. However, both examples are not exciting for the main text. For memory CD8 T cells (Figure 5), there is no way that memory T cells have 5 days lifespan, so obviously the ES model is incorrect (assuming that all labeled cells come from the source/precursors). I think a better example here would be data on neutrophils and illustrating/repeating previous results showing how lifespan depends on the model would be more informative. (And it would be good for authors to comment on their Blood paper in Discussion). As for Ahmed et al 2020 paper data, I think a better analysis would be to assume different from the paper differentiation pathway and see if lifespan estimates depend on it. For example, there is a cell from which both 57+ and 57- cells are made at different rates -> how will this impact estimates of turnover rates? Authors may want to come up with other models that are not assumed in Ahmed et al. Panel B/C in Figure 6 are not useful as the model clearly does not fit the data (showing AIC would be useful).

Limitations. I did not see a part of discussion exclusively devoted to limitations of this work. Please address.

Rewording novelty of the work. From my understanding of the literature, inability of simple models to discriminate between proliferation and source (the latter is precursors in the model) has been shown before (e.g., 9469816, 10799860, 12737664). Hence, focus of some previous work on "closed" populations (no source), e.g., memory T cells. In some ways, it is obvious since from labeling curves that give 2 slopes one can only estimate 2 parameters, and the model with 3 (source, division and death rates) will overfit such data. Also, the debate about neutrophil labeling clearly established that the model for source impacts estimates of cell lifespan. What authors do is to address this point in a more general framework, rather just for neutrophils. That's fine but the abstract and text should be positioned from that perspective, by putting results in the proper context (and not overselling them).

Minor comments

I don't like commenting on nomenclature but I wonder if having name for POI as being separate, rather than N_2 or N (in the model with 1 population). This could also allow to have n precursor populations. But of course, this is just a suggestion.

I feel that most of derivations related to models could be moved to supplement, most results could be stated in the text with giving one formula results or focusing on the figures. Or perhaps moving main derivations into Materials and methods (currently missing from the paper).

When authors talk about kinetic heterogeneity model and just consider 2 populations with different turnover rates (eqn 3), I think authors are imprecise. This is a specific case of "explicit kinetic heterogeneity" model with n=2 populations, the full model in fact includes multiple sub-populations (infinite in the limiting case). Kinetic heterogeneity model should be referred to Asquith at el model with p and d* rates for increase and decline ( 12464572). Claiming that ES and KH models are identical is incorrect in general unless authors show this for KH model with variable (infinite?) number of sub-populations.

I understand the desire to make estimates dimensionless but in some cases it is hard to connect results to other estimates we have for lymphocyte turnover. Would it be possible to have Toff time and then have all rates in per day units? This will make the paper less mathematical and better for biological audience -> this is PCB journal not BMB or JTB.

Figures should be revised to ensure that they can be interpreted by a color-blind individual and when printed on B&W printer (as I did).

Captions for some figures lack details and titles are not very informative. For example, Figure 1 first sentence in bold should give the message of the figure. And also refer to relevant equations in the text. Table 2 caption is poor. Figures 3 and 4 should be squares with the same range for x and y, so the slope can be visually interpreted. Figure 5 caption is poor -> what are these data from previous paper? Are you suggesting I go and read that paper? I don't have the time. Explain what experiment was here. Also, I cannot see well panel letters a) b) c) for panels, make them bigger and bold. In b), the fit is not adequate (model does not fit some data) -> why/? Figure 6 should have SSR or (better) AIC to indicate the fit quality -> obviously in b/c model does not fit the data -> why do we can about these fits?

More detail should be given in how the models were fitted to data, i.e., was the labeling assumed to be step function? Is that a good assumption?

I wonder if knowing the lifespan/turnover of a cell population (e.g., memory T cells), one could make an estimate of the source when fitting s+p+d model to data. Perhaps this is something to be discussed.

Line 702 - I don't think I ever have seen delay in labeling data, Figure 6 or 7 do not show it. So, while the model may predict such dynamics, it is not observed in typical experiments. Perhaps this is something to discuss.

723-724 - isn't that obvious? Seems that it is not a new result really and should be stated as such.

Figure S6 - what is the usefulness of SSR here? Are you comparing alternative models?

Reviewer #2: Swain, Borghans and de Boer have made a very nice contribution to the modeling literature illustrating the issues of trying to infer population turnover rates using cell labeling studies when there is also differentiation of cell types, particularly when slower proliferating cell types entering the pool of interest.

The major conclusion is that if the population one is investigating is not isolated (i.e. has rapid inflow), then estimates could be extremely skewed by rates in the precursor population (i.e. see d1 vs p* in Fig 3). This is a really well written and thought out article, and I just have a couple suggestions of potentially how to make it more broadly read, because I would love if others at least had an intuition of these concerns. Sincerely,

Daniel Reeves

First, it could be nice to provide a sort of executive level summary, ideally written for experimentalists, as to how worried/skeptical they should be about various results. I like your case studies and I think if you had a table/figure with some real numbers, for instance with CD4 or CD8 T cell proliferation/differentiation rates, you might be able to give a traffic signal type simplified (fine, ok, bad) message to describe what kinds of cell subsets are most likely to be mis-estimated, and which other ones are likely to be good enough.

Another useful thing you might do is to explain precisely what extra measurements would really help in cases where misestimation is likely, for instance, you say: "Uniquely identifying the turnover rate of a POI, therefore, requires measurements (or knowledge) on the contribution of label from its precursors" -- does that require labeling and sorting? or are there other types of data that are a little easier to get that might be good enough to ensure estimates are reasonable (e.g. flow data https://journals.plos.org/ploscompbiol/article?id=10.1371/journal.pcbi.1005417)

Lastly, this was a comprehensive look at a particular nuance of labeling, though it left me wondering how two other factors, a) the dynamics of the free label, i.e. what if not a square pulse step function, and b) what about differentiation outward?, and/or what if inward and outward differentiation are balanced? If you do not want to explicitly model these considerations it would at least be useful to mention them in the discussion

Minor:

-might be worthwhile to explain that deuterium is sometimes notated as in heavy water D2O

-I was a little confused where intro ended and results began

-in Fig1 and 2, I think a simple cartoon of the model could help me understand the vectors better, but I still don't quite understand how this works (especially backwards yellow)

-Fig5b doesn't fit so well, do you have any explanation for this? Is there differentiation outward?

-"Deuterium labelling is hailed as the current best technique to accurately measure the turnover rates of a cell population in vivo", probably needs a reference, or references simply to this being a common tool, and again the same point of any insight you can provide to what other tools could help parse out situations where labelling struggles

-very minor, appreciate you sharing all your code, though could definitely benefit from a bit more commenting if someone else is to use

-in this work (https://www.nature.com/articles/s41467-023-41521-1), we used D2O data measured in 5 CD4 subsets to estimate proliferation and differentiation rates of HIV infected cells, however, we relied on the original D20 rate estimates naively, reasoning that differentiation was slower than proliferation so ok to ignore, this might be another useful study to comment on, as I'd be very curious how much our results would have changed if we instead used your modeling approach.

-I wasn't able to see the companion paper, but sounds interesting

Reviewer #3: The manuscript by Swain et al. presents a comprehensive study aimed at exploring how the estimated turnover rate of the deuterium-labeled population of interest (POI) may be affected by the dynamics of its precursor population. The striking examples in Figures 1b and 2b show that the fraction of deuterium-labeled for the POI may increase even when no cell division occurs and when the deuterium is washed out. The manuscript is well-written, and I have only a few minor comments below.

Figure 6: Please report the estimated parameters for Panels (b) and (c), add the value for ‘k’ in the Figure 6 legend, and add equations for the 3 populations in Panel (a) to the supplement.

In Figure 6, the turnover rate of the POI (CD57+CD4+ population) is estimated as d2=0.4/day, which would correspond to a lifespan of about 2 days for these cells. Does this value make sense? Some discussion of the immunological relevance of the estimate would be useful. It would be interesting to see how parameter d2 (and lifespan) depends on the value of ‘k’ when fitting the same dataset.

As it is impossible in many cases to distinguish the fitting between k=1 and k=2 (e.g., Figure 1a,d), it is not clear under which conditions there is a need to use k=2. Please explain.

Line 215: It is clear that Equation (8) does not follow from Equation 1c as stated in the manuscript "the labeling curve of a POI can be written as (from Equation 1c):”, because Equation (8) is more general. Please correct the statement.

Line 416: Should the reference to Swain et al. be added by mistake? It seems out of place.

There are multiple references to the paper by Yan et al., which is problematic since it appears to be under review and inaccessible to reviewers or readers. It might be better to make the current manuscript a standalone paper. Alternatively, based on the provided mention of Yan's manuscript, combining the current manuscript with Yan's paper would be beneficial for the reader, as Yan's paper extends the current framework to the higher value of k=2^i reflecting clonal expansion, which likely results in a relationship similar to the one between k=1 and k=2, only covering a smaller subset of fittings possible for k=1.

**Have the authors made all data and (if applicable) computational code underlying the findings in their manuscript fully available?**

Reviewer #1: Yes

Reviewer #2: Yes

Reviewer #3: Yes

PLOS authors have the option to publish the peer review history of their article (what does this mean? ). If published, this will include your full peer review and any attached files.

**Do you want your identity to be public for this peer review?** For information about this choice, including consent withdrawal, please see our Privacy Policy .

Reviewer #1: No

Reviewer #2: **Yes: ** Daniel Reeves

Reviewer #3: No
---

## [Decision Letter · Decision Letter 1]

27 Jan 2025

PCOMPBIOL-D-24-00934R1

How are estimated cellular turnover rates influenced by the dynamics of a source population?

PLOS Computational Biology

Dear Dr. Swain,

Thank you for submitting your revised manuscript to PLOS Computational Biology. The reviewers in general appreciated the revisions and improvements that were made, and were in general positive. However, there are still some questions remaining or raised by some of the new results. Most of these seem to be addressable by better explanations, more details or discussion. Therefore, we invite you to submit a revised version of the manuscript that addresses the points raised during the review process.

Please submit your revised manuscript within 60 days Mar 29 2025 11:59PM. If you will need more time than this to complete your revisions, please reply to this message or contact the journal office at ploscompbiol@plos.org. Please include the following items when submitting your revised manuscript:

We look forward to receiving your revised manuscript.

Kind regards,

Ruy M. Ribeiro

Academic Editor

PLOS Computational Biology

Amber Smith

Section Editor

PLOS Computational Biology

**Reviewers' comments:**

Reviewer's Responses to Questions

Reviewer #1: Comments

Knowing immediate precursors. I am convinced by your arguments that knowing dynamics of labeling of immediate precursors is important -> it makes sense if s=s(t) in IS model, it will be important. What I found misleading is many statements about that point but only a short paragraph in Discussion that we may not know immediate precursors. I think this has to be stated better, including in the abstract, that we actually may not know for 100% immediate precursors for ANY POI. I would challenge you to give an example where immediate precursor is known 100%. An issue also may be that someone publishes a paper stating that population X differentiates into population Y but then few years later this is shown to be wrong (with some intermediate population or even that X and Y are not really related). To me, there is no solution here except of when one analyzes labeling data, two extreme models should be tried -> fully closed population and fully sourced population (and perhaps with time-dependent source). True turnover rate will be in between estimates provided by the two models. Perhaps that's a useful point in your Discussion.

Comparing explicit kinetic heterogeneity vs. explicit source models. I found the description of this point (lines 154-178) poor. I think what you are trying to say is that if one assumes closed population (no source) -- as explicit kinetic heterogeneity model by Ganusov et al. assumed -- and then consider population with a source, the models will give identical predictions. Thus, by model fit one cannot tell that the population is "closed" or has "source". You should state that explicitly in the beginning. But to be critical here, it is no brainer and I think previous models showed similar point (some of the early De Boer's papers on BrdU). Another point here (as I just realized) that the only true difference between IS and ES models is that in IS model source rate sigma is constant and in ES model it is time-dependent and explicit. However, if I use splines (or similar) to describe time-dependent sigma in IS model (e.g., CD57- cells in the data you analyzed assuming pathway CD57- -> CD57+), then this model would be nearly equivalent to the ES model. Please discuss.

Populations with a source or no source. I think it is critical to discuss situations in which POI can be closed. To me, populations of Ag-specific lymphocytes, e.g., LCMV-specific CD8 T cells, would be closed population (no source) in the absence of reinfection. In this case, we don't need the ES model, explicit kinetic heterogeneity model (with no source) will work fine. Perhaps this is an exception, though.

Peak of labeling. Analyses in supplement clearly show that peaks of labeling in precursors may not always decline (e.g., Fig R3b, R4b), and the data in Fig 5 show higher peak for CD57+ cells than CD57- cells contradicting your model prediction if the model CD57- -> CD57+ is assumed. Experimentally we also expect to see some noise in measurements, so whether peaks are lower (or higher) may be hard to determine in some cases. Also, I wonder if this result of lower peaks in POI as compared to precursors survives non-exponential division times of cells. Finally, I also think that results with slowing accumulating label (D20) shown in the response letter should be put in the supplement.

Analysis of experimental data. I like the new point that estimates of turnover rates of CD57+ T cells depend on the model but the problem here is that to make interpretation one has to assume relationship between CD57+ and CD57- populations. Authors here assume CD57- -> CD57+ pathway but how do you know this is correct? I think that would be a better point of discussion -> not that one has to take into account dynamics of precursors but that how these populations are related and how the rate of conversion determines estimates of the turnover rates. Given that accumulation of CD57+ cells in humans occurs over years most likely these populations are independent on the timescale of labeling, so the model assuming "rapid" differentiation of CD57- to CD57+ may be incorrect (e.g., differentiation may occur once in 10 divisions of CD57- cells). Please address.

Also, regarding the data. Please provide intuitive explanation of how labeling that reached 3% in 50 days (and similar decline of 2% in 50 days) matches 7 day turnover rate you estimate (line 748). This does not make sense to me, turnover rate must be much slower. Perhaps your estimate of 7 days comes from the assumption that labeling lasts 1 day and not 50 days (this is the issue of scaling the models that I pointed out to and that perhaps popped up here as huge error). I would still recommend using T as the duration of labeling and not 1 because I wonder if others may make the same mistake.

I did not see an explicit section on limitations of the paper, limitations appear to be scattered in Discussion. Please put limitations in the specific place/paragraph(s) with appropriate heading/first sentence, e.g., "Our work has following limitations" (or similar) per rigorous science.

Minor comments

Line 146 - I am not sure if eqn. 2h can be fitted to data without knowing k. Then eq. 2h has 3 parameters (p_2, d_2, and k) but we typically think that labeling curves provide estimates for 2, so the model will overfit the data unless k is fixed. Please address.

Lines 15-16 in abstract are confusing. I think the issue is that precursor dynamics have not been taken into account, e.g., source has been constant in previous models. That's the main point of your work, I think (and perhaps having "mechanistic" description of the source).

Line 25 - I think you should add the point that one must know, for 100%, that immediate precursors for POI are actually immediate precursors.

Line 397 -- what if labeling D(t) is not a step function? Then implicit kinetic heterogeneity model could give a delay in accumulation of label in POI.

All figures: please put parameters used for simulations inside each panel. It is hard to go between figure and the caption to see what parameter values are.

In Fig 2 b and d I was surprised by the huge difference in shape of two curves. Is that just because k=1 in b and k=2 in d? Perhaps this is worth noting.

Authors spent a lot of time calculating limits/slopes. These are ok but ultimately nobody should use slopes to estimate rates, we typically fit the whole curve to the data (because estimating slopes may be hard due to noise in the data). Perhaps this is important to mention.

When simulating dynamics of the model (by generating "data"), what was the time step used? How many total datapoints per curve? This info should be provided better but is completely absent now.

I would suggest making Fig 3 and Fig 4 panels squared because ranges on x and y axes are the same. Same point for the "cartoon".

For the cartoon, I am not sure I buy the division into cell types... Perhaps just have some "names" as rapidly dividing POI and small source, slowly dividing POI, or similar. I don't know if we know which areas correspond to which cells (and which species - humans? Mice? Rabbits?)

Table 3 should explicitly show average turnover rates of the cells for different models. Currently it does not. Also, put average rate estimates in Figure 5.

I liked the point that a model with k=2 should not describe the Ahmed et al. data. This could imply that k=1 or that the assumed differentiation pathway is wrong. I think showing model fits with k=2 could be useful -> I.e., how wrong the model predictions are.

First sentence of Figure 5 caption does not match the point of the figure. For example, panel a does not have CD57- population.

Reviewer #2: Thanks for the responses and the work on the danger zone box. I hope this article gets appreciated as I think this is a concept that is relatively intuitive yet mostly ignored and as you show that might lead to large errors.

Reviewer #3: The manuscript has undergone substantial revision and is significantly improved. The much better fit to the Ahmed et al. data in the new Figure 5 and the added cartoon are particularly appreciated. The authors addressed my previous comments, and I have only a few minor points regarding the newly added pieces.

The authors replaced the previous ES model-based Figure 6 with the current Figure 5, where each proposed model includes a kinetically heterogeneous population consisting of two sub-populations. The motivation for this change is described as: “We do this to allow the highest degree of freedom to the model to explain the data.” I found this explanation confusing, as earlier in the text they stated, “the two sub-population kinetic heterogeneity model [6,9] turns out to be a special case of the ES model.”

If I understand correctly, the three-population model in previous Figure 6 is somewhat equivalent to the current two-population model in Figure 5. It may be worth including the case of the ES model with three populations from the previous Figure 6 for comparison, as an alternative model.

For clarity, I strongly suggest adding the corresponding systems of equations used to fit the data in the different panels of new Figure 5 (perhaps in the supplement?).

It appears that the added cartoon is based on a simulation. Please add more details on how it was generated and add the corresponding numbers to the axes.

The following points refer to lines in the revised version with corrections:

• Should the reported values in lines 810 (31%) and 925 (33%) be consistent? Please verify.

• Please add references for lines 959 and 962.

**Have the authors made all data and (if applicable) computational code underlying the findings in their manuscript fully available?**

Reviewer #1: Yes

Reviewer #2: Yes

Reviewer #3: None

PLOS authors have the option to publish the peer review history of their article (what does this mean? ). If published, this will include your full peer review and any attached files.

**Do you want your identity to be public for this peer review?** For information about this choice, including consent withdrawal, please see our Privacy Policy .

Reviewer #1: No

Reviewer #2: **Yes: ** Daniel Reeves

Reviewer #3: No

**Figure resubmission:**
---

## [Editor Report · Decision Letter 2]

14 Apr 2025

Dear Dr. Swain,

The editors appreciate your careful revision taking into consideration the previous round of reviews, and your detailed explanation. Therefore, we are pleased to inform you that your manuscript 'How are estimated cellular turnover rates influenced by the dynamics of a source population?' has been provisionally accepted for publication in PLOS Computational Biology.

Best regards,

Ruy M. Ribeiro

Academic Editor

PLOS Computational Biology

Amber Smith

Section Editor

PLOS Computational Biology

---

## [Editor Report · Acceptance letter]

PCOMPBIOL-D-24-00934R2

How are estimated cellular turnover rates influenced by the dynamics of a source population?

Dear Dr Swain,

I am pleased to inform you that your manuscript has been formally accepted for publication in PLOS Computational Biology. Your manuscript is now with our production department and you will be notified of the publication date in due course.

With kind regards,

Lilla Horvath
